# Marine and freshwater micropearls: Biomineralization producing strontium-rich amorphous calcium carbonate inclusions is widespread in the genus *Tetraselmis* (Chlorophyta)

Agathe Martignier[1], Montserrat Filella[2], Kilian Pollok[3], Michael Melkonian[4], Michael Bensimon[5], François Barja[6], Falko Langenhorst[3], Jean-Michel Jaquet[1], Daniel Ariztegui[1]

[1]Department of Earth Sciences, University of Geneva, Geneva, 1205, Switzerland
[2]Department F.-A. Forel, University of Geneva, Geneva, 1205, Switzerland
[3]Institute of Geosciences, Friedrich Schiller University Jena, Jena, 07745, Germany
[4]Botany Department, Cologne Biocenter, University of Cologne, Cologne, 50674, Germany
[5]EPFL ENAC IIE GR-CEL IsoTraceLab, EPFL, Lausanne, 1015 Switzerland
[6]Microbiology Unit, University of Geneva, Geneva, 1205, Switzerland

*Correspondence to*: Agathe Martignier (agathe.martignier@unige.ch)

**Abstract.**

Unicellular algae play important roles in the biogeochemical cycles of numerous elements, particularly through the biomineralization capacity of certain species (e.g. coccolithophores greatly contributing to the "organic carbon pump" of the oceans) and unidentified actors of these cycles are still being discovered. This is the case of the unicellular alga *Tetraselmis*

*cordiformis* (Chlorophyta) that was recently discovered to form intracellular mineral inclusions, called micropearls, which had been previously overlooked. These intracellular inclusions of hydrated amorphous calcium carbonates (ACC) were first described in Lake Geneva (Switzerland) and are the result of a novel biomineralization process. The genus *Tetraselmis* includes more than 30 species that have been widely studied since the description of the type species in 1878.

The present study shows that many other *Tetraselmis* species share this biomineralization capacity: 10 species out of the 12

tested contained micropearls, including *T. chui*, *T.convolutae*, *T.levis*, *T. subcordiformis*, *T. suecica* and *T. tetrathele*. Our results indicate that micropearls are not randomly distributed inside the *Tetraselmis* cells, but are located preferentially under the plasma membrane and seem to form a definite pattern, which differs between species. In *Tetraselmis* cells, the biomineralization process seems to systematically start with a rod-shaped nucleus and results in an enrichment of the micropearls in Sr over Ca (the Sr/Ca ratio is more than 200 times higher in the micropearls than in the surrounding water or

growth medium). This concentrating capacity varies among species, and may be of interest for possible bioremediation techniques regarding radioactive [90]Sr water pollution.

The *Tetraselmis* species forming micropearls live in various habitats, indicating that this novel biomineralization process takes place in different environments (marine, brackish and freshwater) and is therefore a widespread phenomenon.

## 1 Introduction

The biogeochemical cycles of numerous elements are influenced by the biomineralization capacities of certain unicellular organisms. This is the case, for example, of the coccolithophores, which play an important role in the carbon cycle through their production of biogenic calcite (Bolton et al., 2016). Amorphous calcium carbonate (ACC) is also an important actor in the biogenic carbonate cycle because it is a frequent precursor of calcite, as many organisms use ACC to build bio-minerals with superior properties (Albéric et al., 2018; Rodriguez-Blanco et al. 2017). For example, the precipitation of calcium

carbonate in microbial mats, the Earth's earliest ecosystem, starts with an amorphous calcite gel (Dupraz et al., 2009), and the formation of ACC inside tissue could make coral skeletons less susceptible to ocean acidification (Mass et al., 2017).

In unicellular organisms, intracellular inclusions of ACC had, at first, only been described in cyanobacteria (Couradeau et al., 2012; Benzerara et al., 2014; Blondeau et al., 2018). More recently, similar inclusions have been described in unicellular eukaryotes of Lake Geneva (Switzerland). Consisting of hydrated amorphous calcium carbonates (ACC) but frequently

enriched in alkaline-earth elements (*e.g*. Sr or Ba) and typically displaying internal oscillatory zonation, these inclusions have been named micropearls (Jaquet et al., 2013; Martignier et al., 2017).The internal zonation is due to variations of the Ba/Ca or Sr/Ca ratios.

Until now, micropearls had been observed only in two freshwater species: the unicellular green alga *Tetraselmis cordiformis* (Chlorodendrophyceae, Chlorophyta) producing micropearls enriched in Sr and a second freshwater microorganism producing micropearls enriched in Ba, yet to be identified (Martignier et al., 2017). Since its first description in 1878 (Stein, 1878), the genus *Tetraselmis* has been much studied by biologists, because several species are economically important due to their high

nutritional value and ease of culture (Hemaiswarya et al., 2011). *Tetraselmis* species are used extensively as aquaculture feed (Azma et al., 2011; Lu et al., 2017; Park and Hur, 2000; Zittelli et al., 2006) and some have been suggested as potential producers of biofuels (Asinari di San Marzano et al., 1981; Grierson et al., 2012; Lim et al., 2012; Montero et al., 2011; Wei et al., 2015). They have also served as models in algal research (Douglas, 1983; Gooday, 1970; Kirst, 1977; Marin et al., 1993; Melkonian, 1979; Norris et al., 1980; Regan, 1988; Salisbury et al., 1984).

The motile cells of *Tetraselmis* have four scale-covered flagella, which emerge from an anterior (or apical) depression of the cell (Manton and Parke, 1965). The *Tetraselmis* genus has a cell wall formation process that is unique among green algae as the cells synthetize small non-mineralized scales in the Golgi apparatus, which are exocytosed through Golgi-derived secretory vesicles to form a solid wall (theca) composed of fused scales (Becker et al., 1994; Domozych, 1984; Manton and Parke, 1965). Regarding their habitat, most *Tetraselmis* species are free-living (planktonic or benthic) (Norris et al., 1980) although

some species live in specialized habitats, for example as endosymbiont in flatworms (Parke and Manton, 1967; Trench, 1979; Venn et al., 2008). *Tetraselmis cordiformis* is presumably the only freshwater species among the 33 species currently accepted taxonomically in the genus *Tetraselmis* (Guiry and Guiry, 2018).

However, mineral inclusions had never been described in these microorganisms until the recent observation of micropearls in *Tetraselmis cordiformis* (Martignier et al., 2017). The fact that this new physiological trait had gone unnoticed is puzzling,

especially as *Tetraselmis cordiformis* is the type species of the genus. This can probably be explained by the translucence of the micropearls under the optical microscope and their great sensitivity to pH variations, leading to their alteration or dissolution during most sample preparation techniques (Martignier et al., 2017).

Interestingly, several *Tetraselmis* species (e.g. *T. subcordiformis*) have been mentioned as potential candidates for radioactive Sr bioremediation due to their high Sr absorption capacities (Fukuda et al., 2014; Li et al., 2006) but the precise process by

which these microorganisms concentrate this element has never been determined.

The present study investigates twelve species of the genus *Tetraselmis*, including the freshwater *Tetraselmis cordiformis*, with the objective of understanding whether the biomineralization process leading to the formation of micropearls is common to the whole genus or is restricted to *T. cordiformis*. Species living in contrasting environments have been selected to evaluate also if the formation of micropearls is linked to their habitat. Each species is represented by one or several strains, obtained

from public algal culture collections. All analyses were carried out on cells sampled from these cultures on the day of their arrival in our laboratory The micropearls were imaged by scanning electron microscopy (SEM), and their composition measured by energy-dispersive X-ray spectroscopy (EDXS). The inner structure and chemical composition of micropearls in three different species were studied by transmission electron microscopy (TEM) on focused ion beam (FIB) cross sections.

## 2 Samples and Methods

### 2.1 Origin of the samples and pre-treatment methods

Culture samples of 12 different *Tetraselmis* species were obtained from three different algal culture collections and were grown in different media (Table 1). The recipe of each growth medium is available on the website of the respective culture collections (Table S1). A single strain of each species was studied, except for *T. chui* (2 strains) and *T. cordiformis* (3 strains). Table 1 lists the strain names. Most cells in these cultures were mature at the time of observation for this study.

Samples for microscopic observation of each strain were prepared directly after the organisms' arrival in our laboratory: small portions of the culture (without any change of the original medium) were filtered under moderate vacuum (-20 to -30 kPa) on polycarbonate filter membranes with 0.2, 1 or 2 µm pore sizes. Volumes filtered (variable depending on culture concentration) were recorded. Species issued from SAG (Sammlung von Algenkulturen - University of Göttingen, Germany) were grown on agar and, therefore, cultures had to be resuspended just before filtration. Filter membranes were dried in the dark at room temperature after filtration. A total of 458 micropearls were analysed by EDXS.

### 2.2 Water chemistry measurements

Elemental composition of each culture medium was measured at the IsoTraceLab (EPFL, Lausanne, Switzerland), except for the ES medium for which we could not obtain a sample. Blank samples of MilliQ water were embottled at the same time as growing medium samples and measured in the same way (Table S2). Barium and Sr were measured by Inductively Coupled Plasma Sector Field Mass Spectrometry (ICP-SFMS) using a Finnigan$^{TM}$ Element2 High Performance High Resolution ICP-MS model. The mass resolution was set to 500 to increase analytical sensitivity. Calibration standards were prepared through successive dilutions in cleaned Teflon bottles of 0.1g l$^{-1}$ ICPMS stock solutions (TechLab France). Suprapur® grade nitric acid (65% Merck) was used for the acidification in the preparation of standards. Ultrapure water was produced using Milli-Q® Ultrapure Water System (Millipore, Bedford, USA). Rhodium was used as Internal Standard (IS) for samples and standards to correct signal drift.

At this resolution mode, the sensitivity was better than $1.2 \times 10^6$ cps/ppb of $^{115}$In. The measurement repeatability expressed in terms of relative standard deviation (RSD) was better than 5%. The accuracy of the method was tested using a home-made standard solution containing 5.0 ng l$^{-1}$, used as a reference. Accuracy was better than 5%. The detection limits obtained for Sr and Ba was around 100 ng l$^{-1}$ under these experimental conditions. Note that for the ES medium (not analyzed), the concentrations were set as equivalent to standard sea water, ie. Sr=9 10-5 M. Ca=10-2 M, giving a Ratio Sr/Ca= 9 10-3.

### 2.3 Scanning electron microscopy (SEM) and EDXS analysis

Small portions of the dried filters were mounted on aluminium stubs with double-sided conductive carbon tape and then coated with gold (ca. 10 nm) by low vacuum sputter coating. A JEOL JSM 7001F Scanning Electron Microscope (Department of Earth Sciences, University of Geneva, Switzerland), equipped with an EDXS detector (model EX-94300S4L1Q; JEOL), was

used to perform EDXS analyses and to obtain images of the dried samples. Semi-quantitative results were obtained using the *ZAF* correction method. Samples were imaged with backscattered electrons. This method allows to clearly locate the micropearls inside the organisms, thanks to the high difference of mean atomic numbers between the micropearls and the surrounding organic matter. EDXS measurements were acquired with settings of 15 kV accelerating voltage, a beam current of ~7 nA and acquisition times of 30 seconds. Semi-quantitative EDXS analyses of elemental concentrations were made without taking carbon, nitrogen and oxygen into account. EDXS results are all presented as mol%.

## 2.4 Counts and statistics lead on the *Tetraselmis* culture cells

Counts were performed on the images obtained by SEM. The counts showed that the agar medium seems to hinder the growth of micropearls. These strains were therefore not taken into account for the statistics. Two strains of *Tetraselmis cordiformis* and two strains of *Tetraselmis chui* were analysed. The samples of the two *Tetraselmis cordiformis* strains taken on their first day of arrival were damaged during sample preparation due to a too high filtration pressure, destroying the arrangement of the micropearls in the cells. A sample obtained from one of these strains 60 days after arrival was therefore taken into account for the statistics, in replacement.

The preservation of the pattern of micropearl arrangement in the cell is difficult during sample preparation, as it is easily disturbed. The following parameters directly influence the preservation of that feature: the fragility of the cells (*T. contracta* cells, for example, seem very solid while *T. chui* cells seem more fragile) and sample preparation methods (e.g. pressure during filtration, see difference between (e) and (f) in Fig. S1).

## 2.5 Focused ion beam (FIB) preparation

Electron-transparent lamellae for TEM were prepared with a FIB-SEM workstation (FEI Quanta 3D FEG at the Institute of Geosciences, Friedrich Schiller University Jena, Germany). The cells were previously selected based on SEM imaging. To protect the sample, a platinum strap of 15 to 30 μm in length, ~3 μm wide, and ~3 μm high was deposited on the cell during lamella preparation, via ion-beam induced deposition using the Gas Injection System (GIS). Stepped trenches were prepared on both sides of the Pt straps by Ga+ ion beam sputtering. This operation was performed at 30 keV energy and 3 to 5 nA beam current.

The resulting lamellae were then thinned to approximately 1 μm thickness by using sequentially lower beam currents at 30 keV energy (starting at 1 nA and ending at 0.5 or 0.3 nA). The position of the lamellae was chosen to include a maximum of micropearl cross-sections. An internal micromanipulator with tungsten needle was used to lift-out the pre-thinned lamellae and to transfer them to a copper grid.

Final thinning of the sample to electron transparency (~100 to 200 nm) was carried out on both sides of the lamellae by using sequentially lower beam currents (300 to 50 pA at 30 keV energy). The lamellae underwent only grazing incidence of the ion beam at this stage of the preparation. This allows to minimize ion beam damage and surface implantation of Ga. The thinning

progress was observed with SEM imaging of the lamellae at 52°. Electron beam damage was further supressed by using low electron currents and limiting electron imaging to a strict minimum.

## 2.6 Transmission electron microscopy (TEM) and EDXS analysis

TEM investigations were conducted with a FEI Tecnai $G^2$ FEG transmission electron microscope operating at 200kV. In order to document the structural state of micropearls in their pristine undamaged form, selected-area electron diffraction (SAED) patterns were taken directly at the beginning of the TEM session with a broad beam. Scanning TEM (STEM) images were then acquired using a High Angle Annular Dark Field (HAADF) STEM detector (Fischione) with a camera length of 80 mm. EDXS measurements were performed with a X-MaxN 80T SDD EDXS system (Oxford). EDXS spectra and maps were recorded in scanning TEM mode. The semi-quantitative calculation of the concentrations (including C) was obtained using the Cliff-Lorimer method using pre-calibrated $k$-factors and an absorption correction integrated into the Oxford software. The absorption correction is based on the principle of electroneutrality, taking into account the valence states and concentrations of cations and oxygen anions. Oxygen is thereby assumed to possess a stoichiometric concentration.

## 3 Results and interpretation

TEM analyses confirmed that the mineral inclusions observed in the *Tetraselmis* species during this study comply with the definition of micropearls given in Martignier et al. (2017) (intracellular inclusions of hydrated ACC, frequently enriched in alkaline-earth elements (*e.g.* Sr or Ba) and typically displaying internal concentric zonation linked to elemental ratio variations). These mineral inclusions will therefore be named "micropearls" hereafter.

### 3.1 SEM observation of micropearls in *Tetraselmis* species

SEM observations of twelve different species of *Tetraselmis* (culture strains), on the day of their arrival from the supplier, show that ten of them contained micropearls (Fig. 1, Table 1). None were observed in *T. ascus* and *T. marina*. The general shape of the micropearls in the marine species is elongated, resembling rice grains (Fig. 1 except 1d), while it is spherical in *T. cordiformis* (the only freshwater species of this study) (Fig. 1d). The micropearls' size (0.4-1. 2 μm in length) and shape differ among species. Detailed values for each species are given in Table 1.

Micropearls do not seem to be randomly distributed inside the cells, but rather show a definite location in most species (Figs 1 and S2). Moreover, for a given species, most cells present a similar micropearl arrangement (Fig. S1). Exceptions are cells that were damaged during sample preparation. Filtration or freshwater rinsing, for example, can disrupt the micropearl distribution pattern (Fig. S1-(e) and (f) and Fig. S3).

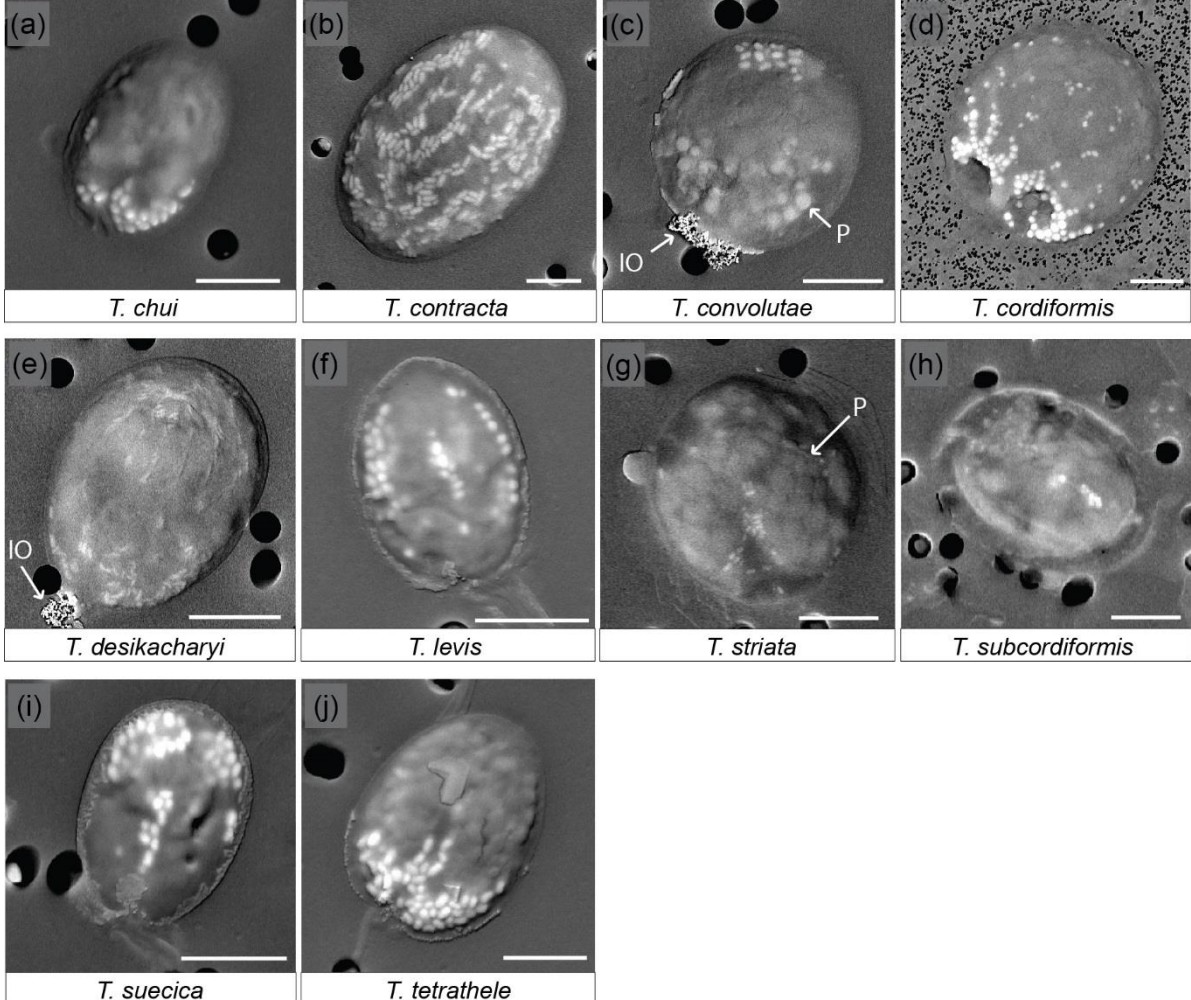

**Figure 1: SEM images of ten *Tetraselmis* species containing micropearls at the time of observation.**

Backscattered electron images of dried samples. The micropearls appear in white or light grey against the darker organic matter, as elongated shapes, except for *T. cordiformis* (d), where they are spherical. P: the larger and slightly darker inclusions are polyphosphates (c, g). IO: iron oxides. Pores of the filters are visible as black circles in the background (2 μm of diameter except for (d): 0.2 μm). Strains: (a): chui_cc; (d): cord-M_cc. Scale bars: 5μm.

In some species, the micropearls are mostly aggregated at one side of the cell, with "pointed" tips appearing at the center of the cell and on both sides, resulting in a "trident" shape. This is the case of *T. chui*, *T. suecica* and *T. tetrathele* (Figs 1a, 1i, 1j). *T. striata* shows a similar central micropearl distribution, but the lateral points of the "trident" are absent (Fig. 1g). In *T. suecica* the central micropearl alignment is generally longer and not necessarily connected to the apical aggregate (Fig. 1i). *T. levis* (Fig. 1f) also shows a similar arrangement, but the aggregate is missing, leaving the micropearls to form three longitudinal alignments (meridians). Altogether, *T. chui*, *T. levis*, *T. suecica* and *T. tetrathele* present patterns with an approximately similar

trimerous radial organization (although a tetramerous symmetry cannot be totally excluded as dried samples do not allow a definite judgement). Observations seem to indicate that, in most species, the micropearl aggregate is located at the apical side of the cell (near the apical depression from which the four flagella emerge) (Fig. S2). In *T. convolutae* (Fig. 1c), the micropearls form a small aggregate at the basal extremity of the cell, while larger polyphosphate inclusions gather at the opposite (apical) side.

A different and interesting organization of the micropearls is displayed by both *T. desikacharyi* (Fig. 1e) and *T. contracta* (Fig. 1b). An apical aggregate of micropearls is generally present, while other micropearls form regularly spaced meridians, which, in *T. contracta*, extend from the apical pole towards the basal part of the cell (Figs 1b and S2). These meridians are not well expressed in all cells but, when they are clearly visible, there seems to be around eight or ten of them inside the cell. When well preserved, the micropearl organization in *T. cordiformis* also shows multiple micropearl alignments which depart from a well-developed apical aggregate, although the alignments are generally well arranged only close to the aggregate and the size of micropearls decreases quickly towards the basal end of the cell (Fig. 1d). Finally, samples observed in this study do not allow us to state if there is a definite distribution of the micropearls in *T. subcordiformis* (Figs 1h and S2).

Polyphosphate inclusions are frequently observed in *Tetraselmis* species. Their distribution seems to be random except in *T. convolutae* (Fig. 1c). Aggregates of small iron oxide minerals were frequently observed in dried samples at one extremity of *T. desikacharyi* and *T. convolutae* (Figs 1c and 1e) – probably at the apical extremity. EDXS analyses performed in both polyphosphate inclusions and iron oxydes aggregates are shown in Fig. S4.

In order to compare our results with members of another genus, we also analyzed other flagellate species (e.g. *Chlamydomonas reinhardtii* and *Chlamydomonas intermedia*) obtained from algal culture collections (Table 1). No calcium carbonate inclusions were observed in these cells. Thorough observation of samples from Lake Geneva confirms that not all flagellates produce micropearls. This biomineralization process seems to be exclusive to a limited number of species.

Table 2 shows the result of counts carried out on the species producing micropearls. On average, 77% of the cells contained micropearls and amongst these, 51% showed the pattern that is characteristic of their species. This last value is high, considering that all cells do not fall onto the filter with the same orientation and that the only patterns we consider are those obtained when the cell is deposited on its lateral side. Patterns resulting of a deposition of the cells on their apical or basal sides are not considered because the 3D repartition of the micropearls in the cells is still uncertain.

### 3.2 TEM observation of FIB-cut cross-sections of micropearls

FIB-cut cross-sections of micropearls produced by *T. chui* and *T. suecica* are shown in Fig. 2, where they are compared to a similar section in a cell of the freshwater species *T. cordiformis* sampled in a natural environment (Lake Geneva). The choice of *T. chui* and *T. suecica* for FIB-processing and TEM observation was based on the size of the micropearls and on their strong concentration in Sr. Both features were considered to favour the observation of compositional zonation, as observed in our previous study (Martignier et al., 2017). A FIB-cut was also performed in a *Tetraselmis contracta* cell. This result is shown

separately in Fig. 3, because the very good conservation of the organic matter in this sample allows the simultaneous observation of other intracellular constituents.

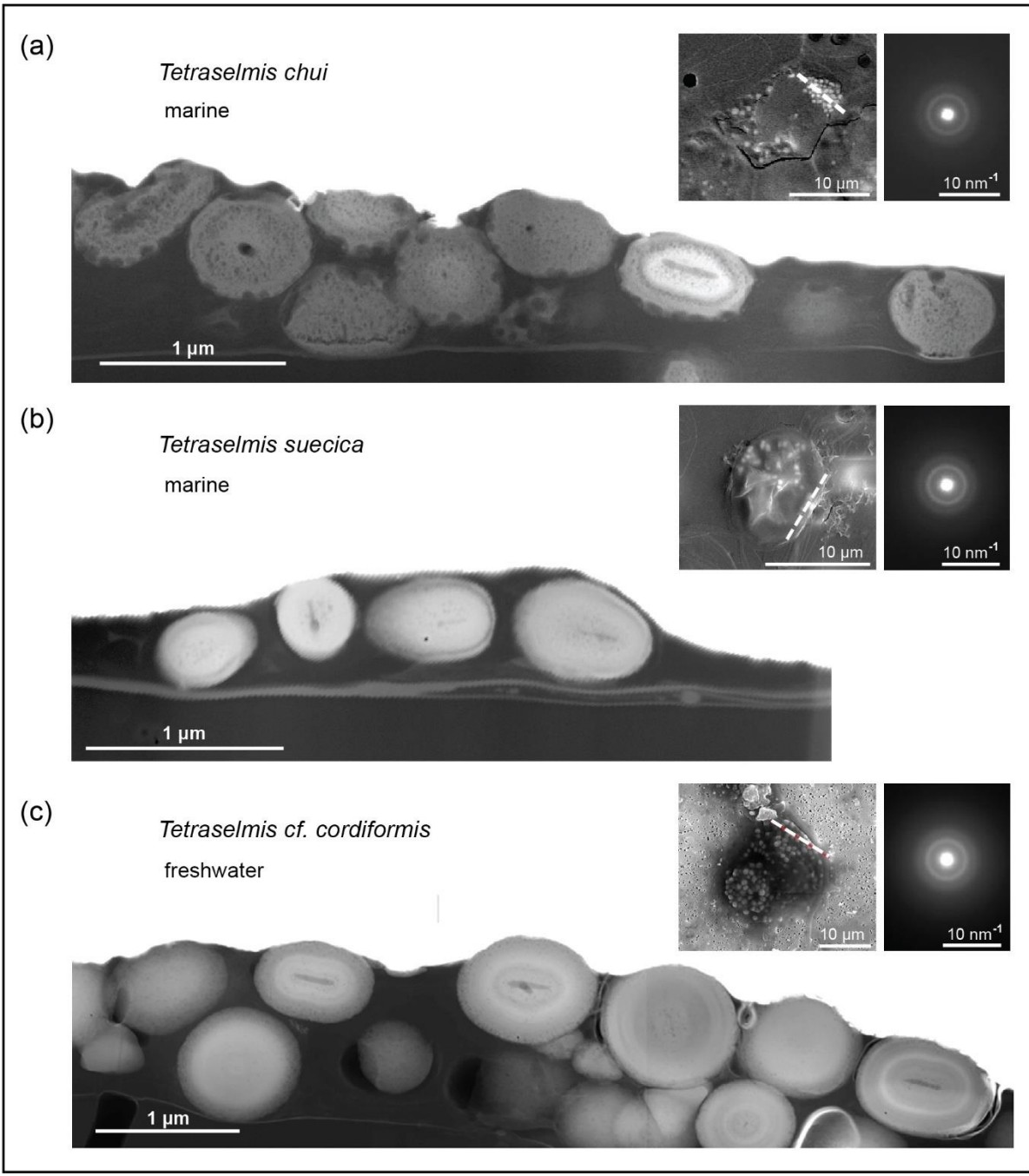

 **Figure 2: Comparison of FIB-cut sections of cells of three different *Tetraselmis* species (dried samples).**

TEM-HAADF images: FIB-cut sections through cells of (a) *Tetraselmis chui* (culture sample); (b) *Tetraselmis suecica* (culture sample); (c) *Tetraselmis cf. cordiformis* (Lake Geneva) (Martignier et al., 2017). Small bubbles inside the micropearls (particularly visible in the marine species) are due to beam damage. The contact between the cell and the filter surface is visible near the bottom in each image. Left top insets: SEM secondary images of the whole cell before FIB preparation indicating the location of the cut with a red line. Right top insets: SAED

patterns from a single micropearls of each FIB-cut section (broad diffraction rings are indicative of amorphous material).

Micropearls in all four species show strong similarities. They are located inside the organic envelope, are amorphous (Figs 2 and 3) and, except for the sample with pure Ca (*T. contracta* in Fig. 3), they show a distinct internal concentric zonation (Fig 2). In all observed species, the cut sections of micropearls suggest the presence of a rod-shaped nucleus in their center (Figs 2

and S5).

As already pointed out, the micropearls are extremely sensitive to the action of the electron beam (Martignier et al., 2017), indicating a vaporization of some of its components: either organic matter associated with water, water contained in the amorphous calcium carbonate (Rodriguez-Blanco et al., 2008), or both. This ACC seems to be rather stable, as beam sensitivity persists after more than five months of storage of dry samples at room temperature.

TEM-EDXS analyses show that the zonation observed in the marine micropearls of *T. chui* and *T. suecica* (Figs 2 and S6) is due to changes in the Sr/Ca concentration ratios, similar to the zonation observed in the freshwater micropearls in *Tetraselmis cf. cordiformis* (Martignier et al., 2017). All micropearls within one cell do not necessarily have an identical composition. An example is shown in Fig. 2a, where one micropearl possesses a composition with a higher atomic mass than the rest (lighter grey level in STEM-HAADF image) due to a higher content of Sr. Furthermore, micropearls within one cell display variable

zoning patterns, as thickness and intensity of the zones differ (Figs 2a and 2c).

### 3.3 TEM-EDXS mapping: location of the micropearls inside a *Tetraselmis contracta* cell

The co-existence of micropearls with other cellular constituents and their respective positions in the cell are shown by a TEM image of a FIB-cut section through a *T. contracta* cell (Fig. 3). The micropearls of this species are large, numerous and nearly exclusively consist of ACC without detectable Sr (Fig. S6). They appear as round to ovoid light grey shapes with smooth

surfaces (Fig. 3a). TEM observations also reveal that most micropearls are not randomly scattered throughout the cell but are located preferentially just under the cell wall.

Although Fig. 3a is difficult to interpret because of the atypical preparation of the sample (simply dried instead of more traditional preparations for TEM-observation such as chemical fixation or cryo-sections), the identification of the visible cellular constituents can still be attempted (Fig. 3b and S7). Side views (lower part of the section) and tangential sections of

starch grains (upper part of the section) are visible, as well as a glancing view of the chloroplast, which is reticulated in this species. Although micropearls resemble starch grains at first look, it is quite easy to differentiate them. First, they are generally more rounded than starch grains; secondly, they are not located inside the chloroplast, in particular, they are not associated with the prominent pyrenoid.

**(a)**

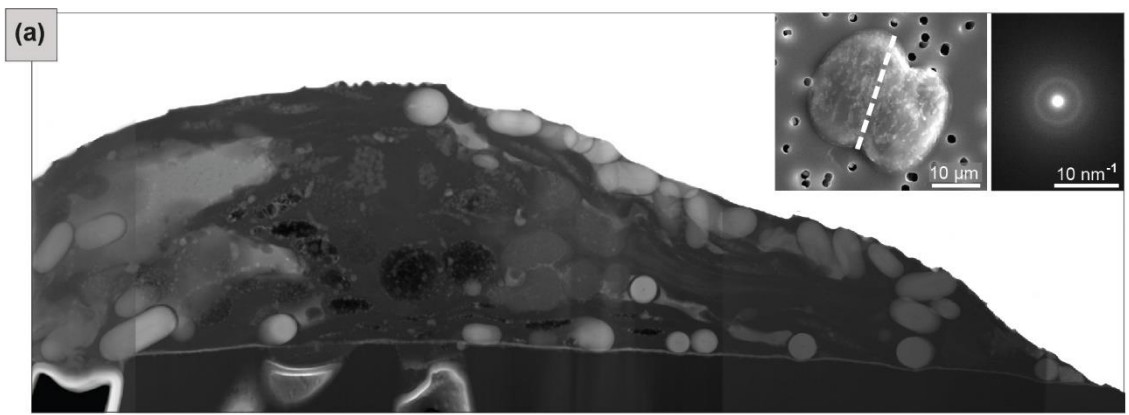

**(b)**

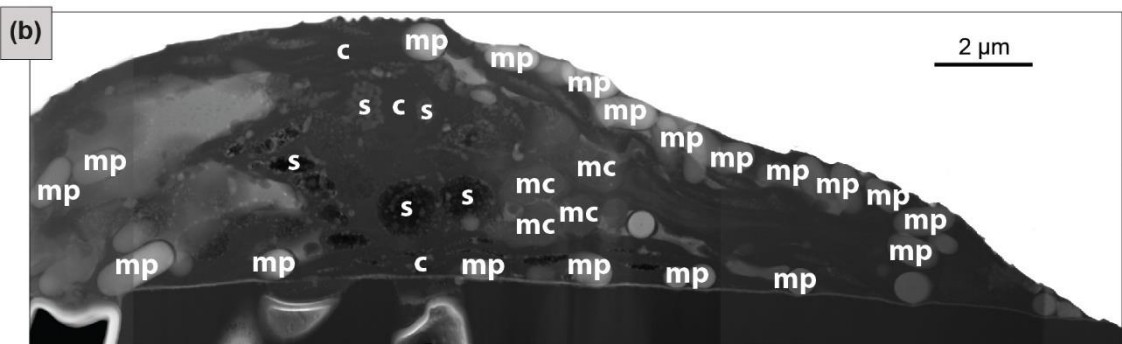

2 µm

c
mp
mp
mp
s c s
mp
mp
mp
s
mc
mp
mp
mc mp mp mp
mc mc mp
mp
mp mp
s s
c mp mp mp mp
mp

**(c)**

flattened PolyP
micropearls

Ca
K
P

µm

**Figure 3: FIB-cut section through a *Tetraselmis contracta* cell (dried sample).**

(a) TEM-HAADF image of the whole FIB-cut section. The micropearls show light or medium grey shades, regular round or oval shapes. Left top inset: SEM secondary images of the whole cell before it was cut, with a red line indicating the location of the section. Right top inset: SAED patterns from a single micropearl of this FIB-cut section (diffuse diffraction rings are indicative for amorphous material). (b) Tentative identification of the visible cellular constituents. s: starch grains; c: chloroplast; mp: micropearls; mc: mitochondria. See Fig. S7 for a detailed image. (c) TEM-EDXS mappings - Top image shows the location of the two zones on a TEM-HAADF image of the section. The map shows an RGB image with three superimposed element mappings. Micropearls are mainly composed of Ca, with small quantities of K (and Mg, not shown here). Note that, due the overlap between the P $K$ peak and secondary Pt $L$ peak, the Pt layer, which was deposited on top of the sample during FIB preparation, is also visible in green color.

TEM-EDXS mapping provides compositional information improving the identification of the cellular constituents and organelles visible in the section (Fig. 3c and S8). Micropearls are well visible, based on the high concentration of Ca, with small quantities of K (and sometimes Mg, not shown here). The theca, composed of fused scales, appears as a thin layer between the cell and the filter. Its composition including C, Ca, S and small amounts of K makes it apparent in Fig. 3c (in violet). The theca of these organisms is indeed known to contain 4% of Ca and 6% of S (as sulfate) by weight (Becker et al., 1994, 1998).

The two irregular features that are highly enriched in P (in green in Fig. 3c) are identified as being PolyP inclusions, flattened during sample preparation. Finally, the dark grey features, in the center of the section, are probably mitochondrial profiles.

### 3.4 SEM-EDXS analysis of micropearl composition

The micropearls of most marine species (Fig. 4a) are composed of ACC, with Ca and Sr as cations. This composition is similar to that measured for micropearls of *T. cordiformis* in Lake Geneva (Martignier et al., 2017). We noted two differences with our previous observations: *T. desikacharyi* forms micropearls containing small amounts of Ba and micropearls of *T. contracta* contain low concentrations of K. However, since growth media had different compositions, these differences need to be taken with care.

Figure 4a compiles the composition of the micropearls for each *Tetraselmis* strain (SEM-EDXS analyses), ranked in increasing order of Sr/Ca median values. Even if low concentrations of K are present in micropearls of *T. contracta,* it was not considered because this element is also present in the surrounding organic matter (Fig. S8), making it impossible to estimate the portion of the measured K that belongs to the micropearls. Magnesium was discarded for the same reason. It should be noted that the size of micropearls is close to or even below the resolution limit of the SEM-EDXS analysis technique. This means that the interaction volume of the electron beam with the sample is often larger than the micropearls themselves. Therefore the technique yields compositions that include the micropearl and the surrounding organic matter or nearby cellular constituents (e.g. polyphosphates).

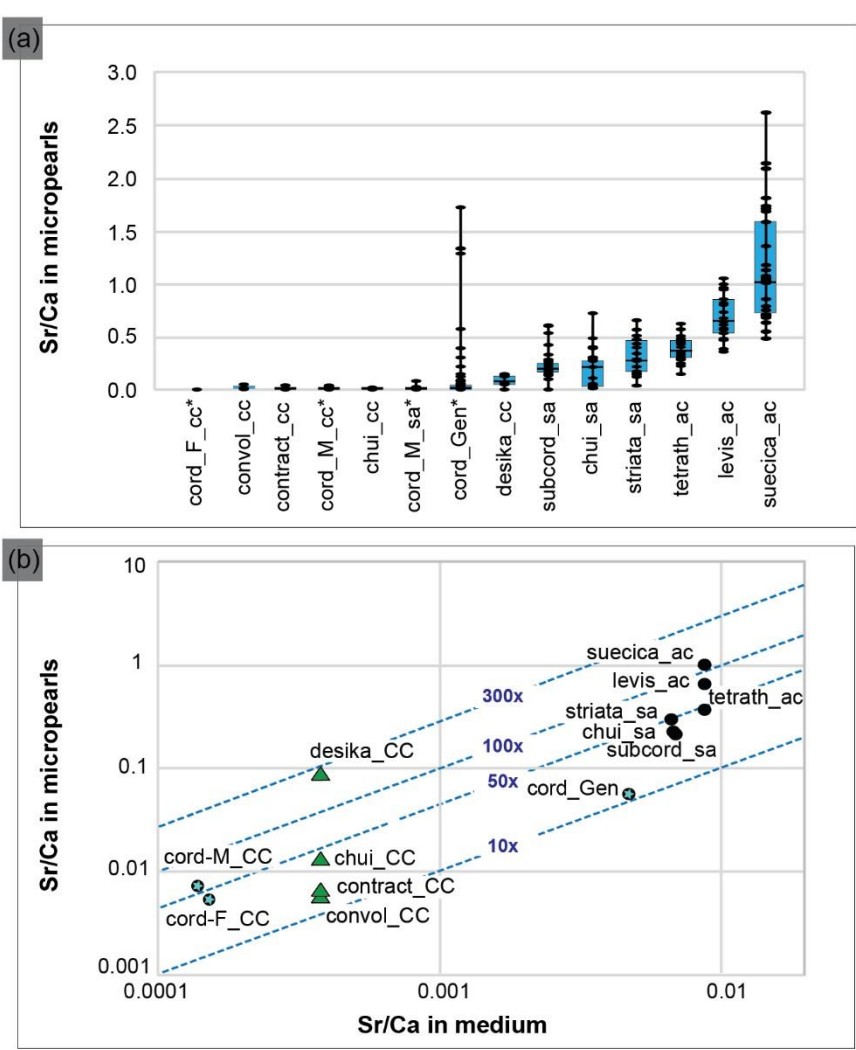

**Figure 4: Composition of the *Tetraselmis* micropearls and their relation with the growth media composition.**

(a) Distribution of the Sr/Ca ratio for each *Tetraselmis* strain (EDXS analyses), ranked according to the median value of Sr/Ca. At least 20 SEM-EDXS analyses were performed on micropearls of each strain. Asterisks highlight freshwater strains. The range between the minimum and maximum data are shown by black lines. The blue boxes represent the 25-75% interquartiles, while the black horizontal line in the boxes shows the median value. (b) Relationship between the composition of the growth media and the composition of the *Tetraselmis* micropearls, expressed as the Sr/Ca ratio. Each point represents the median Sr/Ca ratio measured in each species micropearls, related to the Sr/Ca ratio of the growth medium. Points with blue stars highlight freshwater strains. The blue dotted lines define the values of the Sr enrichment factor of the micropearls with respect to the medium (10x, 50x, etc.). Calcium concentrations of the growth media were calculated, based on media theoretical composition. Green triangles signal four samples grown in the same medium. The abbreviations and characteristics of each strain are indicated in Table 1 while Sr/Ca values appear in Table S2 (for medium) and S3 (for micropearls). Results from *T. cordiformis* from Lake Geneva (cord_Gen) (Martignier et al., 2017) are given as a comparison.

### 3.5 ICP-SFMS analysis of Sr/Ca ratio in growth media: data and interpretation

The concentration of Sr and Ba in the culture media are given in Table S2 and represented graphically in Fig. S9. Strontium concentrations range from $3.3 \times 10^{-8}$ M (freshwater medium SFM) to $7.1 \times 10^{-5}$ M (seawater SWES medium). All media have lower Sr concentrations than the average seawater ($9.1 \times 10^{-5}$ M). SFM, used to grow *T. cordiformis* - the only freshwater strain under study – has lower Sr concentrations than those measured in Lake Geneva ($5.2 \times 10^{-6}$ M).

The molar ratio Sr/Ca has been calculated for seven growth media (Table S2) and 458 micropearls (Table S3) in order to evaluate a possible influence of the medium on the micropearls composition. Differences between the species regarding the micropearls enrichment in Sr compared to their growth medium can be observed. A Sr distribution coefficient (or enrichment factor) was calculated as the molar ratio [(Sr micropearls / Ca micropearls) / (Sr medium / Ca medium)]. Figure 4b shows the relationship between the Sr/Ca ratio measured in the growth media and in the *Tetraselmis* micropearls. For most of the strains, the Sr enrichment factor of the micropearls with respect to the medium varies between 10 and 100 times (see Table S3 for exact figures), with the notable exception of *T. desikacharyi* (more than 200 times). It is interesting to observe that both strains of *T. chui* - from different geographic origins (Table 1) - have rather similar Sr distribution coefficients (around 30), while the three strains of *T. cordiformis* show slightly different enrichment factors (25 for Lake Geneva water, 33 for Lake Fühlingen and 51 for Münster castle moat). Broadly speaking, Sr/Ca increases in micropearls together with its increase in the medium. However, the spread in enrichment may be large for a given medium (such as ASP-H for strains of *T.contracta*, *T. convolutae*, *T. chui* and *T. desikacharyi*).

### 4 Discussion

Micropearls had been previously interpreted as a feature specifically related to freshwater environments (Martignier et al., 2017). The present results show that the biomineralization process leading to the formation of micropearls can take place in very different environments. The following paragraphs aim to discuss our present knowledge on micropearls in general, on their formation process, as well as the newly discovered widespread biomineralization capacity in the *Tetraselmis* genus, involving high concentration capacities of these organisms regarding Sr.

### 4.1 Marine and freshwater micropearls

The discovery of micropearls in marine species of *Tetraselmis* shows that this biomineralization process can take place in organisms living in waters of different composition, from freshwater, like Lake Geneva, to seawater (Fig. S9). This highlights the capacity of these organisms to integrate Ca and Sr from different external media.

The production of micropearls is clearly not directly related to a specific habitat, since seven *Tetraselmis* species forming micropearls live as phytoplankton in freshwater, marine or brackish waters (Guiry and Guiry, 2018; John et al., 2002), *T.*

*contracta* and *T. desikacharyi* were sampled in the sand, at the bottom of a marine estuary (Marin et al., 1996) or at low tide, and *T. convolutae* is usually observed as a photosymbiont inside a flatworm (Muscatine et al., 1974). Regarding the only two species which did not show micropearls at the time of observation (*T. ascus* and *T. marina*), it is interesting to note that both live as stalked sessile colonies, with motile life-history stages (Norris et al., 1980).

Apart from their elongated shape, "marine" micropearls have characteristics similar to micropearls formed by the freshwater species *T. cordiformis* (Martignier et al., 2017). Micropearls show a range of possible composition for each species (Fig. 4a and Table S3). The Sr/Ca ratio seems to be influenced by several parameters, amongst which we identified the composition of the culture medium (Fig. 4b) and the Sr concentrating capacity of each *Tetraselmis* species (e.g. green triangles in Fig.4b). Indeed, the general trend seen in this diagram is an adaptation of the ACC precipitation to the medium composition. However,

more relevant information is provided by the enrichment factor (*E factor,* see Table S4 and dotted isolines in Fig. 4b), which allows to rank species (Table S4) from low values (12-16) to more than 200. This ranking would need to be confirmed by cultivating the species in different media (eg. *T. convolutae* group in ES and *T. tetrathele* group in ASP-H) and comparing the new enrichment factor with the current values. The very high *E factor* for *desikacharyi* can tentatively be linked to distinctive morphological features (a six-layered theca, a novel flagellar hair subtype) not found in other strains of *Tetraselmis* (Marin et

al., 1996).

The pattern drawn by the arrangement of the micropearls in the cell is clearly more homogeneous within a strain as compared to between strains. Statistics show that these patterns are characteristic for a given species (Table 2 and Fig. S1), which means that the organisms probably can exert a strong control on the number, size and organization of the micropearls in the cells.

### 4.2 Hints about the formation process of micropearls

The biomineralization process leading to the formation of micropearls seems to start in the same way in all *Tetraselmis* species observed in FIB sections (*T. chui, T. contracta, T. cordiformis* and *T. suecica*), with a similar rod-shaped nucleus (Figs 2, 3 and S5). These nuclei could possibly be of organic nature given their darker appearance in the STEM- HAADF images that point to a material of lower atomic mass (Fig. S5).

It is important to note that there are many parameters which seem to influence the presence/absence of micropearls in the cells:

the state of the culture (fully healthy or suffering from the transport, for example), the *pH* of the medium and probably other parameters we are not yet aware of. For example, the use of agar as culture medium seems to hinder the development of micropearls (Table 2 and Fig. 1g and 1h). Nevertheless, the composition of the medium does not seem to influence the arrangement of the micropearls in the cell, as demonstrated by *T. chui*, *T. contracta* and *T. convolutae* (respectively Fig. 1a, 1c and 1d), which have different patterns, although all were cultured in ASP-H medium.

Internal concentric zones are observed in the micropearls formed by cells grown both in the natural environment and in cultures (Fig. 2). The presence of this concentric pattern, even when the growth media have a stable composition, may indicate that the zonation is not due to changes in the surrounding water/medium composition during micropearl growth, but rather depends on variations in the intracellular fluid composition caused by the biomineralization process itself. In the hypothesis discussed by

Thien et al. (2017), it is suggested that the formation of the micropearls results from a combination of a biologically controlled process (preferential intake of specific cations inside the cell) and abiotic physical and chemical mechanisms (mineralization resulting from a non-equilibrium solid-solution growth mechanism, leading to an internal oscillatory zoning). Nevertheless, even that second part of the process does not seem to be purely abiotic, as demonstrated by the long-term amorphous state displayed by micropearls (at least five months, according to our observations). Indeed, synthetic ACC with no additives is unstable and rapidly crystallizes into calcite or aragonite (Addadi et al., 2003; Bots et al., 2012; Weiner and Addadi, 2011, Purgstaller, 2016), often through the intermediate form of vaterite (Rodriguez-Blanco et al., 2011). In contrast, long-term stabilization of ACC implies the presence of mineral or organic additives (Aizenberg et al., 2002, Sun et al. 2016). Magnesium is known to play a key role in the stabilization of ACC by introducing a distortion in the host mineral structure (Politi et al., 2010). This might well be the case for the *Tetraselmis*-hosted micropearls, in which Mg content is around 2 mol%. Although the phosphate ion has also been reported to inhibit ACC crystallization (Albéric et al., 2018), it does not seem to be the case here, the phosphorus concentration of the micropearls being below the detection level of EDXS. Stabilization of ACC is also enhanced by certain proteins, polyphosphonates, citrates and amino acids (Levi-Kalisman et al., 2002; Addadi et al., 2003; Cam et al., 2015; Cartwright et al., 2012). The presence of these molecules inside the micropearls is suggested by their observed sensitivity to beam damage. As for the possible role of Sr in the ACC long-term stability, we did not find in the literature any reference thereof. However, in an *in vitro* experiment, Littlewood et al. (2017) found, in the presence of Mg, a correlation between added Sr and the reaction time to transform ACC into calcite (2 h to a maximum of 24 h).

**4.3 A new intracellular feature in a well-known genus**

Our results (Fig. 1) confirm that artefacts can be induced by usual biological sample preparation techniques (Martignier et al., 2017) and thus introduce biais in observations and even hide some physiological traits in otherwise well-studied organisms. Figure 3c shows that the straightforward sample preparation method used in this study (dried, with no chemical fixation) allows the preservation of the micropearls and yields useful data on the composition of the different elements present inside the cell, without any chemical disturbance.

Micropearls represent a new intracellular feature. Their systematic presence in most of the analysed *Tetraselmis* species suggests that they probably play a physiological role. A possible explanation could be that micropearls increase the sedimentation rate of cells that shed their flagella upon N starvation at the end of *Tetraselmis* blooms. An alternative hypothesis is that micropearls represent reserves of Ca for periods when millimolar Ca is not available in the external medium. Indeed, most Chlorodendrophyceae are known to require the presence of $Ca^{++}$ to survive and multiply (Melkonian, 1982). The evolutionary diversification of this class occurs in the marine habitat, where the Ca concentration is constantly around 10 mM (Table 4.1 in Pilson, 1998). The need for Ca is supported by *T. cordiformis*, the only freshwater species of the genus, occurring only in Ca-rich lakes, with a minimum of 1 mM of Ca (e.g. Lake Geneva (1 mM) or Fühlinger See (2mM)), and tests on cultures showed that *T. cordiformis* cannot develop normally in an environment with 0.42 mM of Ca (Melkonian, 1982). Calcium is needed to support phototaxis (light-oriented movements) and for the construction and maintenance of the cell

coverage (theca, flagellar scales) (Becker et al., 1994; Halldal, 1957). The Sr found in the composition of the micropearls formed by most *Tetraselmis* spp. (Fig. 4) could be transported by the same transporter as Ca. Indeed, *Chlorodendrophyceae* have very efficient light-gated Ca-channels (channelrhodopsins) which are also essential for phototaxis of these flagellates (Govorunova et al., 2013; Halldal, 1957).

**4.4 Bioremediation possibilities**

The capacity of some organisms to concentrate Sr is of great interest regarding bioremediation. Strontium ($^{90}$Sr) is one of the radioactive nuclides released in large quantities by accidents such as Chernobyl or Fukushima (Casacuberta et al., 2013) and a major contaminant in wastewater and sludge linked with nuclear activities (Bradley et al., 1996). Its relatively long half-life of ~30 years and high water solubility cause persistent water pollution (Thorpe et al., 2012; Yablokov et al., 2009). For

example, the desmid green alga *Closterium moniliferum*, which can incorporate 45 mol% of Sr in barite crystals, is considered as a potential candidate as a bioremediation agent (Krejci et al., 2011). The high Sr absorption capacity of several *Tetraselmis* species also led to their mention as potential candidates for radioactive Sr bioremediation (Fukuda et al., 2014; Li et al., 2006). In our experiments, *T. suecica*, for instance, produced a high number of micropearls which contained more than 50 mol% of Sr when cultured in ES medium (data not shown). Nevertheless, the process allowing these microorganisms to concentrate Sr

had not yet been investigated and further studies of micropearl formation processes could therefore lead to new bioremediation techniques. The genus *Tetraselmis* presents the additional advantage of including species living in diverse habitats, which might offer interesting bioremediation applications in different aquatic environments including freshwater, brackish lakes, open sea and hypersaline lagoons (Table 1).

**5 Conclusions**

Until recently, non-skeletal intracellular inclusions of calcium carbonate were considered nonexistent in unicellular eukaryotes (Raven and Knoll, 2010). After the first observation of at least two micropearl-forming organisms in Lake Geneva (Martignier et al., 2017), the present study shows that these amorphous calcium carbonate (ACC) inclusions are widespread in a common phytoplankton genus (*Tetraselmis*), not only in freshwater, but also in seawater and brackish environments. This newly discovered biomineralization process therefore takes place in media of very different composition and our results suggest that

it is similar in all studied species: an oscillatory zoning process that starts from an organic rod-shaped nucleus. Although frequent in this well-studied genus, these mineral inclusions had been overlooked in the past, possibly destroyed by the usual sample preparation techniques for electron microscopy. Thus other microorganisms could have similar capacities and intracellular inclusions of amorphous calcium carbonates may be more widespread than currently known.

Micropearls represent a new intracellular feature. This study shows that they can be clearly distinguished from other cellular

constituents and are not randomly distributed in the cell. On the contrary, micropearls seem to be essentially located just under

the cell wall and they draw a pattern which suggest to be characteristic for each species. Strong correlations hint that this might have a link with the species habitat.

It appears that, for most of the observed *Tetraselmis* species, the biomineralization process leading to the formation of micropearls enables a selective concentration of Sr.. The elements concentrated in the micropearls, as well as their degree of enrichment seem to be characteristic for each species. Selecting the species with the highest concentration capacities could be of high interest for bioremediation, especially regarding radioactive Sr contaminations linked with nuclear activities.

## Author contribution

AM designed and lead the study, conducted the SEM and EDXS analysis, analysed EDXS results and SEM images and wrote the paper. MF was a key collaborator for writing the article, provided expertise and key contacts. JMJ and MF helped design the study. JMJ carried out most sample preparations and processed EDXS data. KP produced the FIB-sections, conducted the TEM analysis and processed the results. MM interpreted the TEM images of the *T. contracta* cross section and provided biological expertise as a specialist of the genus *Tetraselmis*. MB led the ICPM-MS analyses of the growth media. FB provided key contacts and biological advice. FL contributed to theFIB-TEM study. DA is AM's thesis supervisor; he helped design the study, provided expertise and funded the project. All authors discussed the results and commented on the manuscript.

## Competing interests

The authors declare that they have no conflict of interest.

## Acknowledgements

This research was supported by the Société Académique de Genève (Requête 2017/66) and the Ernst and Lucie Schmidheiny Foundation. We thank Mauro Tonolla, Sophie de Respinis and Andreas Bruder (SUPSI) as our collaboration triggered the present research, Barbara Melkonian and the CCAC (University of Cologne) for their help and collaboration, and Maike Lorenz (SAG- University of Göttingen) for culture tips and allowing the analysis of their growth medium. We also thank Stephan Jacquet and Andrew Putnis for advice, as well as Rossana Martini and Camille Thomas for support. The critical and constructive comments of the three reviewers are gratefully acknowledged. FL is grateful to the Deutsche Forschungsgemeinschaft for funding of the FIB-TEM facilities via the Gottfried-Wilhelm Leibniz program (LA830/14-1).

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

|  | origin of the sample | approx. micropearl size | culture medium | provider strain n° | abbreviation |
|---|---|---|---|---|---|
| **Chlamydomonas** | | | | | |
| **C. reinhardtii** | **freshwater** France | no micropearls observed | L-C | TCC 778 | - |
| **C. Intermedia** | **freshwater** France, Lake Geneva | no micropearls observed | L-C | TCC 113 | - |
| **Tetraselmis** | | | | | |
| **T. ascus** | **marine** Spain, Canary Islands, St Cristobal | no micropearls observed | ASP-12 | CCAC 3902 | - |
| **T. chui** | **marine** Germany, Heligoland | 0.7 µm length | ASP-H | CCAC 0014 | chui_cc |
|  | **marine** Scotland, Millport, Clyde estuary | 0.7 µm length | 1/2 SWEg Ag | SAG 8-6 | chui_sa |
| **T. contracta** | **marine** France, Bretagne, Batz island | 1.2 µm length | ASP-H | CCAC 1405 | contract_cc |
| **T. convolutae** | **marine** (symbiotic in flatworm) France, Bretagne, Batz island | 0.8 µm length | ASP-H | CCAC 0100 | convol_cc |
| **T. cordiformis** | **freshwater** Germany, Cologne, Lake Fühlinger | 1 µm diameter | SFM | CCAC 0051 | cord-F_cc |
|  | **freshwater** Germany, Münster, castle ditch | 1 µm diameter | Waris - H | CCAC 0579B | cord-M_cc |
|  | **freshwater** Strain 0579B obtained from CCAC | 1 µm diameter | Diat | SAG 26.82 | cord-M_sa |
| **T. desikacharyi** | **marine** France, Batz island, Rochigou | 0.9 µm length | ASP-H | CCAC 0029 | desika_cc |
| **T. levis** | **marine** France, Saint-Gilles-Croix-de-Vie | 0.6 µm length | ES | AC 257 | levis_ac |
| **T. marina** | **marine** Strain CA5, from L. Provasoli | no micropearls observed | Porph Ag | SAG 202.8 | - |
| **T. striata** | **marine** UK, N-Wales, Conway | 0.6 µm length | SWES Ag | SAG 41.85 | striata_sa |
| **T. subcordiformis** | **marine** USA, Connecticut, New Haven | 0.4 µm length | Porph Ag | SAG 161-1a | subcord_sa |
| **T. suecica** | **marine** UK, Plymouth | 0.7 µm length | ES | AC 254 | suecica_ac |
| **T. tetrathele** | **marine** - | 0.9 µm length | ES | AC 261 | tetrath_ac |

**Table 1: Specific information for each species and their culture conditions.**

Providers: CCAC: Culture Collection of Algae at the University of Cologne (Germany) - www.ccac.uni-koeln.de ; SAG: Sammlung von Algenkulturen at the University of Göttingen (Germany) - https://www.uni-goettingen.de/de/184982.html ; AC: Algobank - culture collection of microalgea of the University of Caen (France) - www.unicaen.fr/algobank; TCC: Thonon Culture Collection of the CARRTEL of Thonon-les-Bains (France) - www6.inra.fr/carrtel-collection. All culture media compositions are given on the corresponding websites (detailed addresses in Table S1).

| *Tetraselmis* | strain | medium | total cells counted | % cells with mp /cells | % pattern / cells with mp | Remarks |
|---|---|---|---|---|---|---|
| *T. chui* | CCAC 0014 | ASP-H | 160 | 93 | 40 | |
| *T. chui* | SAG 8-6 | 1/2 SWEg Ag | 121 | 40 | 37 | resuspended from agar |
| *T. contracta* | CCAC 1405 | ASP-H | 103 | 98 | 79 | |
| *T. convolutae* | CCAC 0100 | ASP-H | 100 | 40 | 70 | |
| *T. cordiformis* | CCAC 0051 | SFM | 115 | 60 | 0 | strongly filtered |
| *T. cordiformis* * | CCAC 0579B | Waris-H | 123 | 98 | 46 | gently filtered |
| *T. desikacharyi* | CCAC 0029 | ASP-H | 122 | 25 | 13 | |
| *T. levis* | AC 257 | ES | 123 | 94 | 51 | |
| *T. striata* | SAG 41.85 | SWES (agar) | 136 | 12 | 25 | resuspended from agar |
| *T. subcordiformis* | SAG 161-1a | Porph (agar) | 100 | 1 | 0 | resuspended from agar |
| *T. suecica* | AC 254 | ES | 105 | 99 | 57 | |
| *T. tetrathele* | AC 261 | ES | 101 | 89 | 56 | |

**Table 2: Percentage of cells presenting micropearls and specific patterns of micropearl arrangement**

Percentage of cells presenting micropearls for each strain and percentage of these cells showing the typical micropearl arrangement pattern for that species (see Figs 1 and S1). Two strains have been analysed for *T. chui* and *T. cordiformis*. Please note that strains grown on agar generally show a much lower presence of micropearls and were not considered for the statistics. The asterisk marks a single sample taken 60 days after the strain's arrival in our laboratory, while all the others were observed on the first day after arrival from the provider. This exception allowed to estimate the number of cells showing the micropearl arrangement pattern of this species, as both samples of *T. cordiformis* strains taken on the first day were damaged during sample preparation by a too strong filtration. On the first day after arrival, strain CCAC0579B gave results similar to those of strain CCAC 0051. mp = micropearls. For details on providers and media, see Table 1.