# Peer review of "Marine and freshwater micropearls: Biomineralization producing strontium-rich amorphous calcium carbonate inclusions is widespread in the genus *Tetraselmis* (Chlorophyta)"

_Biogeosciences, 2018_

## Referee Comment (RC1) · E. Couradeau (Referee) · 23 May 2018

I read with great interest the manuscript entitled "Marine and freshwater micropearls: Biomineralization producing strontium-rich amorphous calcium carbonate inclusions is widespread in the genus Tetraselmis (Chlorophyta)" by Martignier et al.

This paper reports the observation of micropearls, which are intracellular amorphous carbonate formed by unicellular eukaryotes in 14 samples (out of the 16 samples examined) encompassing 11 strains of the genius Tetraselmis. The samples were obtained from culture collections and cells were dehydrated upon arrival on a membrane filter to be further observed using SEM coupled to elementary X-EDS analyses. Some FIB sections were also prepared and analyzed by TEM. This piece of work deepens our view of micropearl formation showing that it is not limited to the freshwater T. cordiformis previously found in lake Geneva but also occurs in a large set of marine species. It also shows that the micropearls form in standard culture conditions and that they can express the Sr-Ca zonation pattern in constant culture condition. Interestingly the authors looked at the nucleus of the micropearls and showed that it is a rod shaped organic nucleus suggesting the importance of organic template to initiate the nucleation of the micropearl and to maintain it in an amorphous stage. The authors suggest that the Sr bioremediation properties attributed to the genus Tetraselmis could be linked to their ability to concentrate Sr in mineral.

I found this paper very interesting and well written. It is easy to follow and I don't have any concern that that could preclude its publication in BG.

I have a couple of general comments/suggestions that hopefully will help make this paper an even stronger contribution:

It is unclear to me from reading the manuscript if all the cells from a species had the micropearls. I would like to see some kind of measurement of how many cells had them and if the pattern of biomineralization was more homogenous within a strain as compared to between strains. If it is the case (especially for strains grown in the same media) it would suggest a high level of control of the number/size/organization of the micropearls.

Regarding the 90Sr remediation potential of Tertraselmis, I was wondering if the author could calculate from their estimates of the composition of the micropearls the contribution of the mineral phase to the "Sr absorption capacities of several Tetraselmis" P15 L19. In other words can we quantitatively link the potential of micropearl forming to the Sr absorption?

I was also wondering how these species compared in term of Sr concentration to the one forming Sr and Ba sulfates, that could make an additional interesting point to discuss. See for instance: Krejci, M. R., Wasserman, B., Finney, L., McNulty, I., Legnini, D., Vogt, S., et al. (2011). Selectivity in biomineralization of barium and strontium. J. Struct. Biol. In Press,. Available at: http://www.sciencedirect.com/science/article/pii/S1047847711002346.

Detailed comments:

P2 L4 why are the micropearls "non-skeletal", your data suggest that the micropearls are organized probably along the cytoskeleton; they could be an organizing component of the cell serving as skeleton/internal spatial organization principle.

P2 L20 "two freshwater organisms" > what is the second one?

P3 L29 resuspended instead of "diluted"

P4 L4 Barium and strontium (instead of Sr)

P4 L12 what is RSD?

P4 L20 what is ZAF?

P7 L20 iron oxide are extracellular? are they always in contact with the cell?

P7 L25 to be exclusive TO a limited number

P15 L6 "require a certain concentration of Ca", which is ?

Figure 4 (a) tertrah is not in the Table 1, is it Tetrah_ac or _sa? (b) is the ES medium Enriched Seawater ? If so you could use the S rand Ca concentration of seawater as proxy for this medium as it is composed of filtered seawater amended with metal and vitamins mostly (no addition of Sr & Ca). That would allow you to plot the marine species on the part

(b) of the graph which would be interesting because they have the highest Sr/Ca ratio

in the micropearls.

I will now answer the 15 BG questions:

1. Does the paper address relevant scientific questions within the scope of BG? Yes, the topic of biomineralization is within the scope of BG

2. Does the paper present novel concepts, ideas, tools, or data? Yes the fact that micropearls are widespread beyond the one species already described and also occur in marine environment is new. Although the techniques are not novel, the FIB prep shown in figure 3 is outstanding knowing that the sample was not fixed or preserved in any way.

3. Are substantial conclusions reached? Yes

4. Are the scientific methods and assumptions valid and clearly outlined? Yes

5. Are the results sufficient to support the interpretations and conclusions? Yes

6. Is the description of experiments and calculations sufficiently complete and precise to allow their reproduction by fellow scientists (traceability of results)? Yes

7. Do the authors give proper credit to related work and clearly indicate their own new/original contribution? Yes

8. Does the title clearly reflect the contents of the paper? I found the title too long but indeed it reflects the content of the paper

9. Does the abstract provide a concise and complete summary? Yes

10. Is the overall presentation well structured and clear? Yes

11. Is the language fluent and precise? Yes.

12. Are mathematical formulae, symbols, abbreviations, and units correctly defined and used? Yes

13. Should any parts of the paper (text, formulae, figures, tables) be clarified, reduced, combined, or eliminated? No

14. Are the number and quality of references appropriate? Yes

15. Is the amount and quality of supplementary material appropriate? Yes

―――――――――――――

---

## Referee Comment (RC2) · A. Immenhauser (Referee) · 25 May 2018

Dear Editor, dear Authors,

Thank you for providing me with the opportunity to review this interesting ms on the formation of amorphous calcium carbonate in the cells of micro-algae. Obviously, I am not a lacustrine microbiologist hence my comments are those of a person interested in carbonate biomineralization, metastable carbonate phases, and the role of elemental cycles in aquatic environments. Moreover, I have read the review of E. Couradeau

and my below remarks do not iterate the – in my view – well justified criticism already brought forward. This allows me to place my comments at a higher conceptual level.

Generally, my impression of this ms is very positive. This is modern, state-of-the-art research dealing with a hitherto less than sufficiently well studied topic. From the perspective of a carbonate sedimentologist and a person interested in the interface between abiogenic and biogenic processes, however, I have a few general comments, some of which are, as indicated above, also editorial in nature and aim for a paper that is accessible to a wider readership.

1) I do not think that the abstract serves well to attract the interest of a wider readership outside of the specialized community performing focussed research in this field. Please consider to start the abstract with a topical statement on element cycles and the role of algae in this "game". The immediate focus on Tetraselmis leaves the reader with the impression of a somewhat narrow approach. I think the paper as such is much broader actually and the abstract undersells the significance of this story.

2) The Introduction, albeit often well written, is in part a bit unstructured or so it seems and I would like to see that the authors provide text regarding the aims of this paper. I guess that would be pretty standard and I know that many readers like to have an idea of the general direction the paper takes.

3) The results chapter is generally well-designed but in places transgresses the boundary between genuine data presentation and interpretation, perhaps the most commonly found criticism in reviews these days. That could be rather easily solved by using a header such as Data "Presentation and Interpretation" and by restructuring the text in a manner that physically separates (paragraphs) more descriptive text from more interpretative text. Again, by this I aim for a better accessibility of the text for the non-specialized reader. Clearly, chapter 3.5 is more of an interpretation than a genuine data reporting. Please consider.

4) Discussion: I was – in places – a bit confused about the manner in which the au-

B

thors jump between lacustrine and marine micropearls. Could you do the reader the favour and commence the discussion with a paragraph explaining the reader how you structure the text? After all, the title of this paper refers to lacustrine and marine case examples but I do not see any chapter in the discussion specifically referring to lacustrine micropearls (but there is one dealing with marine examples actually referring to freshwater ones in the first paragraph)? I am aware of the fact that you have published on lacustrine examples before and that you place the lacustrine and marine findings into context but this all seems a bit unbalanced and would clearly benefit from some form of a better structure.

5) Generally, I found the literature cited on ACC somewhat "classical". These are clearly landmark papers worth citing but a series of more recent studies dealing with thermodynamic and kinetic and biomineralization aspects on how and why organisms secrete or induce amorphous phases seems absent or so I think. I am happy to provide references should the authors wish so.

6) The chapter, brief as it might be, that I really miss is one providing the reader with information about the wider significance of the amorphous calcium carbonate with respect to carbonate cycles and elemental cycling in these water bodies. Could you provide back on the envelope estimates about the volumes of material that are cycled here and the temporal constraints (seasonal patterns)?

7) Conclusions: Please don't come up with a paper written in 2010 (Raven and Knoll) and refer to something that was considered non-existent at that time. I agree, eight or so years seem not long ago but in our hyperactive research environment, this is actually a long time and significantly more accurate and recent findings have been published since then.

Summing up: My comments aim to improve an already very nice paper and I clearly encourage the journal Biogeosciences to consider publication of this work. My comments are on a conceptually high level given that a technical review is already available and

given that I would like to see a wider readership making use of the science documented here.

I hope my comments are of use to the authors.

Sincerely yours, Adrian Immenhauser

---

## Referee Comment (RC3) · M. Alberic (Referee) · 27 May 2018

The presence of intracellular amorphous calcium carbonate inclusions (called micro-pearls) has been identified for the first time in marine unicellular micro-algae (genus Tetraselmis). A wide range of marine species has been studied and compared to a fresh water species from the same genus. Careful and high quality structural and chemical investigations were performed, which allow characterizing the main structural features (shape, size, spatial localization of the micropearls and other cellular components) as well as chemical composition (Sr/Ca ratio) of the micropearls for each

species. The results are discussed in terms of biomineralization processes, possible functions of the micropearls are proposed and bioremediation application highlighted. I believe this study is very interesting for the readership of Biogeosciences and the manuscript is very well written. The authors may address the following comments that mainly concern the organization of the results and a more advanced discussion part that could impact more fields as for example the "ACC stabilization" research community. My only small concern is about the timescale of the biomineralization processes. It will strengthen the paper if some scale can be provided.

Detailed comments

Introduction. The first sentence giving the definition of "micropearls" should be precised because this term was first proposed in the last study of the authors (Martignier 2017) for one genus (Tetraselmis). "Intracellular mineral inclusions of ACC" have been identified before in others species (in particular cyanobacteria) and were not called micropearls. Therefore, the genus should be stated and the reference (Martignier et al. 2017) added. In addition, previous studies (Couradeau et al 2012 and later ones Benzerara et al. 2014), should be cited even if they concern prokaryote organisms. Line 20. "two freshwater organisms", could mean either two individuals or two species... Only one is cited (cordiformis), what is the other one?

2.1. The presence of micropearls might depend on the time of observation of the cells. A time scale should be therefore indicated approximately, in order to make sure that it is the same for all culture cells. Would it be possible that micropearls belonging to different species could have different sizes, shapes, spatial localizations, Sr/Ca ratios just because of different time scales and not because they are from different species? Is the rate growth of the micropearls known? Does the compositional zonation (number of lines or spacing) could be a marker of time? or a marker of the different steps in the biomineralization process?

2.3. The coating was gold, therefore the authors should state why carbon, nitrogen and

oxygen where not taken into account in the semi-quantitative analyses.

2.5. Some EDXS have been done so the title should be changed accordingly

3. Results. In general the subtitles are not homogenous. If the authors choose to name the subtitles according to the techniques they use they should be more consistent, and therefore called 3.1 "SEM observation. . ." , and possibly put 3.2 and 3.3 together and call it TEM-EDXS, (because EDXS analysis are also reported in 3.2).

3.1. In Figure 1, the full name of the samples are not reported for d) and j) that have different strains. The different strains might be very similar but this should be specified. In addition in Table 1 there is a mistake in the sample names for cui_sa and chui_cc. Line 11. "strains"or "species"?

p.7 line 1. The "problem" of the time of observation appears here, it is reported just for T. sriata, but the different organization in the different species could not be also related to different time scale? For T. Levis "the aggregate is missing" again, is this related to time?

Polyphosphate inclusions are not easily seen in Figure 1. Higher magnifications would be useful. Or relate to the TEM- EDXS observations? Did the authors observed EDXS signal from the P in these SEM images? Same for Iron-oxide minerals.

Line 22 and 25 "organisms" or "species"?

The authors cannot really state that "most flagellates do not produce micropearls" if they studied only two other species of one genus. They should be more careful, and maybe write instead of "most, . . . do not produce" "not all, . . . produce".

3.2. The choice of the samples for the FIB-sections is not clear. Why T. cordiformis from the culture was not considered? it would have been useful to compare with the natural environment one. It looks like the choice of the species was made in order to observe the compositional zonation. However, the compositional zonation of the different species in barely describe in the paper.

Figure 2. The red line is not visible in a black and white printed version, a dashed white line will be more useful. Higher magnification of individual micropearls like in Martignier et al. 2017 will be useful to better see the compositional zonation.

Fig. S4 shows higher magnifications, but it is difficult to see the zoning pattern. Is it because of the image quality? It looks like T. contracta does not show zoning pattern at all? (in Fig S4,"c)"is missing).

p.9 line 8. "the highly hydrated" state of the ACC should be speculate more carefully. Could it be that the water associated with organic molecules around or within(?) the ACC micropearls could lead to the "strong response under the electron beam"? When dehydration occurs, does ACC eventually crystallize into calcite or is it still stable? Because of the presence of the Sr ?

The unexpected stability of ACC in living systems is still highly debated. And to my knowledge not much studies so far reported the role of Sr on ACC stability, therefore, it will be worth discussing it in the discussion part.

line 14. Describe in which sense (number of lines, spacing. . .) the zoning pattern varies within one cell. Could it be related to different stage of biomineralization within one cell? The same for the different composition within one cell?

3.3. line 22-27. The low magnification of Figure 3a and 3b does not allow properly visualizing the different cellular components, namely starch grains and chloroplasts. On which criteria the authors based the identification of these elements? Structural features? If so, higher magnifications of the areas presenting the chloroplasts and starch grains are needed. Do starch grains and chloroplasts can be characterized by specific chemical elements like PolyP and the scales? What about mitochondrial profiles? Are they also identified according to their shape? Higher magnification is then needed.

3.4. In Fig. 4, it would help the reader to indicate which species are from marine and

fresh environment. Are this data coming from SEM-EDXS or TEM-EDXS or both?

Fig. S5 reports TEM-EDXS, and the section is about SEM-EDXS. No SEM-EDXS figures seem to be reported neither in the text nor in the supplementary info? Or Fig. S5 is a SEM-EDXS figure. Could the authors please clarify this point?

3.5. p13. Line 2. If the "overall composition of all culture media is rather similar" how could one study the influence of the composition of the media on the micropearls composition?

An interesting result is the one line 18, which would I think deserve more discussion in 4.1.

4.Discussion. 4.2. p.14 line 18 to 25. Stabilization of ACC might also be achieved by inorganic ions (phosphates, magnesium, strontium?). The concentration of these ions might still be controlled by the organism. More recent papers about the stabilization of synthetic ACCs in presence of inorganic ions should be cited.The role of Strontium in the stabilization ACC should be further discussed by reporting the literature.

To what refers "ACC in its "pure form"? this is rather vague, the authors could use the term "synthetic ACC with no additives" instead?

Moreover, synthetic ACC even if without inorganic or organic additives can be stable if it is stored in a desiccator for example. ACC in solution crystallizes indeed rapidly but not in air.

4.3. The spatial localization of the micropearls in a cell as well as the differences between cells of different species should be discussed. Does the specific localization close to the flagella have a role in swimming capacities? Center of gravity?

5. Conclusion. p.16 line 6to 9 belongs to discussion not really to the conclusion

BG questions: 1. Yes 2. Yes 3. Yes 4. Yes 5. Yes 6. Yes in general, only one detail might be needed to precise p. 4. L 12. What is RSD what is the "home made

standard solution"? 7. Yes in general, but it would be more precise if they include in the introduction the presence of such intracellular ACC inclusions also in other organisms as cyanobacteria (Couradeau 2012, Benzerara 2014) 8. Yes 9. Yes 10. Yes 11. Yes 12. Yes 13. Some clarification need to be provided (see detailed comments) 14. More recent reference on the stabilization of ACC would be needed 15. Yes

---

## Author Comment (AC1) · 6 Jul 2018

**Reply to reviewer n°1**

**E. Couradeau (Referee)**
ecouradeau@lbl.gov

I read with great interest the manuscript entitled "Marine and freshwater micropearls: Biomineralization producing strontium-rich amorphous calcium carbonate inclusions is widespread in the genus Tetraselmis (Chlorophyta)" by Martignier et al. This paper reports the observation of micropearls, which are intracellular amorphous carbonate formed by unicellular eukaryotes in 14 samples (out of the 16 samples examined) encompassing 11 strains of the genius Tetraselmis. The samples were obtained from culture collections and cells were dehydrated upon arrival on a membrane filter to be further observed using SEM coupled to elementary X-EDS analyses. Some FIB sections were also prepared and analyzed by TEM. This piece of work deepens our view of micropearl formation showing that it is not limited to the freshwater T. cordiformis previously found in lake Geneva but also occurs in a large set of marine species. It also shows that the micropearls form in standard culture conditions and that they can express the Sr-Ca zonation pattern in constant culture condition. Interestingly the authors looked at the nucleus of the micropearls and showed that it is a rod shaped organic nucleus suggesting the importance of organic template to initiate the nucleation of the micropearl and to maintain it in an amorphous stage. The authors suggest that the Sr bioremediation properties attributed to the genus Tetraselmis could be linked to their ability to concentrate Sr in mineral.

I found this paper very interesting and well written. It is easy to follow and I don't have any concern that that could preclude its publication in BG. I have a couple of general comments/suggestions that hopefully will help make this paper an even stronger contribution:

1) It is unclear to me from reading the manuscript if all the cells from a species had the micropearls. I would like to see some kind of measurement of how many cells had them and if the pattern of biomineralization was more homogenous within a strain as compared to between strains. If it is the case (especially for strains grown in the same media) it would suggest a high level of control of the number/size/organization of the micropearls.

   **Authors' response:**

   *We thank the reviewer for this very interesting question. We provide corresponding additional data (Table 2 and Fig. S7). Table 2 will be added in the text and Fig. S7 (part (a) and part (b)) will be added in the supplementary material. We will also include a paragraph in 3.1*

*describing these results and their significance will be shortly discussed in 4.1 or 4.2.*
*Before describing these new results in detail and answering the question of the reviewer, we*
*would like to emphisize that there are many parameters, which we do not yet fully*
*understand and which seem to greatly influence the presence/absence of micropearls in the*
*cells. These are the state of the culture (fully healthy or suffering from the transport, for*
*example), the pH of the medium and probably other parameters we are not yet aware of.*
*Regarding the pattern of biomineralization, it is even more complicated because it seems to*
*be very easily disturbed. So, in addition to the previously mentioned factors, the following*
*ones also influence the conservation of the micropearl pattern: the fragility of the cells needs*
*to be taken into account (T. contracta cells, for example, seem very solid while T. chui cells*
*seem more fragile) and the micropearl pattern can very easily be lost during sample*
*preparation (e.g. too strong vacuum during filtration, see difference between (e) and (f) in*
*Fig. S7). Finally, all cells do not fall onto the filter with the same orientation, and we do not*
*yet fully understand the 3D geometry of these patterns. Nevertheless, we believe that the*
*new results allow us to confirm that there must indeed be a high level of control on the*
*number / size / organization of the micropearls by the cell, as suggested by the reviewer.*

| *Tetraselmis* | strain | medium | total cells counted | % cells with mp /cells | % pattern / cells with mp | Remarks |
|---|---|---|---|---|---|---|
| *T. chui* | CCAC 0014 | ASP-H | 160 | 93 | 40 | |
| *T. chui* | SAG 8-6 | 1/2 SWEg Ag | 121 | 40 | 37 | resuspended from Agar |
| *T. contracta* | CCAC 1405 | ASP-H | 103 | 98 | 79 | |
| *T. convolutae* | CCAC 0100 | ASP-H | 100 | 40 | 70 | |
| *T. cordiformis* | CCAC 0051 | SFM | 115 | 60 | 0 | filtered strongly |
| *T. cordiformis* * | CCAC 0579B | Waris-H | 123 | 98 | 46 | filtered gently |
| *T. desikacharyi* | CCAC 0029 | ASP-H | 122 | 25 | 13 | |
| *T. levis* | AC 257 | ES | 123 | 94 | 51 | |
| *T. striata* | SAG 41.85 | SWES (Agar) | 136 | 12 | 25 | resuspended from Agar |
| *T. subcordiformis* | SAG 161-1a | Porph (Agar) | 100 | 1 | 0 | resuspended from Agar |
| *T. suecica* | AC 254 | ES | 105 | 99 | 57 | |
| *T. tetrathele* | AC 261 | ES | 101 | 89 | 56 | |

**Table 2: Percentage of cells presenting micropearls and specific patterns of micropearl arrangement**

Percentage of cells presenting micropearls for each species and percentage of these cells showing the typical micropearl arrangement pattern for that species (see Fig. 1). Two strains have been analysed for *T. chui* and *T. cordiformis*. Please note that strains kept on Agar generally show a lower presence of micropearls. The asterisk marks a single sample taken 60 days after the strain's arrival in our laboratory, while all the others were observed on the first day after arrival from the provider. This exception aimed to have a representation of a better estimation of the number of cells showing the micropearl arrangement pattern of this species, as both samples of *T. cordiformis* strains taken on the first day were damaged during sample preparation, destroying the arrangements. On the first day after arrival, strain CCAC0579B gave results similar to those of strain CCAC 0051. mp = micropearls. An image of each strain is provided in Fig. S7. For details on providers and medium, see Table 1.

[Figure]

**Fig. S7 (part 1): SEM backscattered overview images of most strains observed in this study**

*(a) T. chui* (CCAC 0014); (b) *T. chui* (SAG 8-6) ; (c) *T. contracta* (CCAC 1405) ; (d) *T. convolutae* (CCAC 0100) ; (e) *T. cordiformis* (CCAC 0051) ; (f) *T. cordiformis* (CCAC 0579B) ; (g) *T. desikacharyi* (CCAC 0029); (h) *T. levis* (AC 257) ; (i) *T. striata* (SAG 41.85) ; (j) *T. subcordiformis* (SAG 161-1a) ; (k) *T. suecica* (AC 254) ; (l) *T. tetrathele* (AC 261).

These images aim to better illustrate the general aspect of the different strains at time of observation. They correspond to the measurements presented in Table 2. Micropearls appear as white dots inside the cells. In the images where only a few cells contain micropearls, white arrows indicate their position. Note that polyphosphate inclusions (e.g. in (d)) or NaCl crystals (e.g. in (j)) can also appear as white dots. Distinction was based on their close-up morphology or EDXS analyses. All images were made on the day following the reception of the strains from the provider, except (f), taken 60 days after reception. This exception aims to show the internal pattern of micropearls in *T. cordiformis*,- pattern that was which was destroyed during our first sample preparation. Note that strains (b), (i) and (j) were maintained on Agar, unlike the other strains. Scale bars: 20 micrometers.

[Figure]

**Fig. S7 (part 2)**

*In summary, these new results show that*
*(1) not all the cells from one species have micropearls.*
*(2) the pattern of micropearl arrangement is clearly more homogeneous within a strain as compared to between strains: the pattern of T. contracta and T. desikacharyi was never observed in other strains. Similarly, the pattern of T. chui or T. suecica was never observed in T. contracta or T. convolutae. These patterns are really characteristic of the species, as shown in the two new figures we provide here. The example of T. chui (a), T. contracta (c) and T. convolutae (d) can be taken, as these three strains have clearly different patterns, and they were all cultured in ASP-H medium. We can note that T. chui, T. levis, T. suecica and T. tetrathele present really close patterns and only between these species can you sometimes*

*observe one cell with the pattern of another species of this same group.*
*(3) The new table (Table 2) provides % of the number of cells containing micropearls in each strain, as well as the number of cells presenting the "micropearl pattern" described in Fig. 1 for this specific species. If we do not take into account the species which were maintained on Agar (which clearly seems to pose a problem for the micropearl production), the average percentage of cells containing micropearls is 77 %. And amongst these cells with micropearls, 51% show the pattern which is characteristic of their species (without taking T. cordiformis CCAC0579B into account, as this sample was damaged during sample preparation).*

2) Regarding the 90Sr remediation potential of Tertraselmis, I was wondering if the author could calculate from their estimates of the composition of the micropearls the contribution of the mineral phase to the "Sr absorption capacities of several Tetraselmis" P15 L19. In other words can we quantitatively link the potential of micropearl forming to the Sr absorption?

**Authors' response:**
*We plan to do so in a follow-up of the present article. Briefly, our approach will imply computing a Sr budget in both soluble and particulate form between time 1 and 2. If we suppose that the decrease in Sr sol during the time interval is due solely to its precipitation within micropearls (to be checked), then we can compare it with an independent estimate of the amount of Sr contained in the micropearls. We are at the moment exploring how to do it based on micropearls counting by analysis of SEM/EDXS images.*

3) I was also wondering how these species compared in term of Sr concentration to the one forming Sr and Ba sulfates, that could make an additional interesting point to discuss. See for instance: Krejci, M. R., Wasserman, B., Finney, L., McNulty, I., Legnini, D., Vogt, S., et al. (2011). Selectivity in biomineralization of barium and strontium. J. Struct. Biol. In Press,. Available at: http://www.sciencedirect.com/science/article/pii/S1047847711002346.

**Authors' response:**
*We will indeed add this information into the discussion part. Krejci et al. (2011) demonstrate the incorporation of 45 mol% Sr  in barite crystals produced by desmid green algae. The micropearls in T. suecica in ES medium reached more than 50 mol% Sr.*
*Reference:*
Krejci, M.R., Finney, L., Vogt, S. and Joester, D. (2011): Selective sequestration of Strontium in Desmid Green Algae by biogenic co-precipitation with barite. ChemSusChem **4**, 470-473

Detailed comments:

4) P2 L4 why are the micropearls "non-skeletal", your data suggest that the micropearls are organized probably along the cytoskeleton; they could be an organizing component of the cell serving as skeleton/internal spatial organization principle.

**Authors' response:**
*We used the term non-skeletal as meaning "not part of a solid skeleton" (such as the frustule*

*for diatoms or the "skeletal-plates" of the coccolithophores). We suggest that we can simply remove this term from the text, as it appears that geologists and biologists understand different things under the same term. Regarding the cytoskeleton, T. cordiformis does not have a microtubular cytoskeleton under its plasma membrane (the "cyto"skeletal function is provided by the cell wall/theca of this organism).*

5) P2 L20 "two freshwater organisms" > what is the second one?

**Authors' response:**
*We thank the reviewer for spotting this lacking information. We will replace the present sentence by the following: "Until now, micropearls had been observed only in two freshwater species: the unicellular green alga Tetraselmis cordiformis (Chlorodendrophyceae, Chlorophyta) producing micropearls enriched in Sr and a second freshwater microorganism producing micropearls enriched in Ba, yet to be identified (Martignier et al., 2017)."*
*For the moment, the only thing we know about this second organism is its approximate size (20 microns) and that it has at least two flagella. We are currently trying different methods to investigate the question (isolation by cytometry, pipette subsampling).*

6) P3 L29 resuspended instead of "diluted"

**Authors' response:**
*We agree with this remark and will implement the suggestion of the reviewer in the text.*

7) P4 L4 Barium and strontium (instead of Sr)

**Authors' response:**
*We agree with this remark and will implement the suggestion of the reviewer in the text.*

8) P4 L12 what is RSD?

**Authors' response:**
*Relative Standard Deviation. We will add this into the text.*

9) P4 L20 what is ZAF?

**Authors' response:**
*The "ZAF correction" is the usual name of the correction method we used during the (semi-) quantification of the EDXS results. This is not an acronym. SEM-EDXS quantification needs to*

*go through a correction taking into account (1) the atomic number effect (Z), (2) the self-absorption effect (A) and (3) the fluorescence effect (F). ZAF correction is one of the standard EDXS correction schemes. We feel that explaining all this in detail would be too long for a "Methods" paragraph. We would therefore prefer to leave this sentence as it is.*

10) P7 L20 iron oxide are extracellular? are they always in contact with the cell?

**Authors' response:**
*Yes, the iron oxides are definitely extracellular. They are most of the time in contact with the cells, although they are also sometimes observed independently. It is impossible to tell if these lone iron oxide aggregates were formed on a cell and then separated during filtration or if they were formed independently.*

11) P7 L25 to be exclusive TO a limited number

**Authors' response:**
*We agree with this remark and will implement the suggestion of the reviewer in the text.*

12) P15 L6 "require a certain concentration of Ca", which is ?

**Authors' response:**
*As explained P15, line 7: "*The need for Ca is supported by *T. cordiformis*, the only freshwater species of the genus, occurring only in Ca-rich lakes, with a minimum of 1 mM of Ca (e.g. Lake Geneva (1mM) or Fühlinger See (2mM))." See also Melkonian, M. (1982) : *in that study, the author state that "survival experiments indicate that most of the cells survive the 15-minute incubation time in calcium-free medium, rod-shaped scales are not regenerated until after formation of new flagella in the next division cycle. It should however be noted that the next division cycle occurs only after a considerable lag phase (several days) in which the cells appear unable to attach to a substratum and divide." The author also determined that when exposed to 0.42 mM Ca2+ the cells lose 50% of their outer flagellar scales suggesting that this may be close to the minimum Calcium concentration tolerated by T. cordiformis.*
*All other Tetraselmis spp. occur in marine/brackish environments where the Ca concentration is about 10mM. For the second genus of Chlorodendrophyceae, the freshwater Schefferlia (Chlorodendrophyceae), no information is currently available on the Ca concentration in its natural environment.*
*We agree to modify this sentence to include more detailed information.*

*Reference:*
*Melkonian, M. (1982) : Effect of divalent cations on flagellar scales in the green flagellate Tetraselmis cordiformis. Protoplasma* **111**, 221-233

13) Figure 4 (a) tertrah is not in the Table 1, is it Tetrah_ac or _sa?

**Authors' response:**

*We thank the reviewer for spotting this mistake. Indeed, this is tetrath_ac. We will apply the corresponding changes to the figure.*

14) Figure 4 (b) is the ES medium Enriched Seawater ? If so you could use the Sr and Ca concentration of seawater as proxy for this medium as it is composed of filtered seawater amended with metal and vitamins mostly (no addition of Sr & Ca). That would allow you to plot the marine species on the part (b) of the graph which would be interesting because they have the highest Sr/Ca ratio in the micropearls.

**Authors' response:**

*We have followed this suggestion, making a Ca and Sr composition equivalence between sea water ($10\text{-}2ML^{-1}$ and $9\ 10\text{-}5ML^{-1}$, respectively, found in literature) and ES medium. This led to the addition of suecica_ac, tetrathele_ac and levis_ac in new Fig. 4b diagram. We will integrate this new version in the next manuscript version.*

---

## Author Comment (AC2) · 6 Jul 2018

**Reply to reviewer n°2**

**A. Immenhauser (Referee)**
adrian.immenhauser@rub.de

Dear Editor, dear Authors,
Thank you for providing me with the opportunity to review this interesting ms on the formation of amorphous calcium carbonate in the cells of micro-algae. Obviously, I am not a lacustrine microbiologist hence my comments are those of a person interested in carbonate biomineralization, metastable carbonate phases, and the role of elemental cycles in aquatic environements. Moreover, I have read the review of E. Couradeau and my below remarks do not iterate the – in my view – well justified criticism already brought forward. This allows me to place my comments at a higher conceptual level. Generally, my impression of this ms is very positive. This is modern, state-of-theart research dealing with a hitherto less than sufficiently well studied topic. From the perspective of a carbonate sedimentologist and a person interested in the interface between abiogenic and biogenic processes, however, I have a few general comments, some of which are, as indicated above, also editorial in nature and aim for a paper that is accessible to a wider readership.

1) I do not think that the abstract serves well to attract the interest of a wider readership outside of the specialized community performing focussed research in this field. Please consider to start the abstract with a topical statement on element cycles and the role of algae in this "game". The immediate focus on Tetraselmis leaves the reader with the impression of a somewhat narrow approach. I think the paper as such is much broader actually and the abstract undersells the significance of this story.

**Authors' response:**
*We thank the reviewer for this suggestion. We will replace the first paragraph of the Abstract by the following text:*
*"Unicellular algae can play important roles in the biogeochemical cycles of numerous elements, particularly through the biomineralization capacity of certain species (e.g. coccolithophores greatly contributing to "organic carbon pump" of the oceans) and unidentified actors of these cycles are still being discovered. This is the case of the unicellular alga Tetraselmis cordiformis (Chlorophyta) that was recently discovered to form intracellular mineral inclusions, called micropearls, which had been previously overlooked. These intracellular inclusions of hydrated amorphous calcium carbonates (ACC) were first described in Lake Geneva (Switzerland) and are the result of a novel biomineralization process. The genus Tetraselmis includes more than 30 species that have been widely studied since the description of the type species in 1878.*
*The present study shows that many other Tetraselmis species share this biomineralization capacity…"*

2) The Introduction, albeit often well written, is in part a bit unstructured or so it seems and I would like to see that the authors provide text regarding the aims of this paper. I guess that would be pretty standard and I know that many readers like to have an idea of the general direction the paper takes.

**Authors' response:**

*We agree with this remark from the reviewer. We will try to give a better structure to the introduction. The last paragraph (Page 3, Line 11) already describes the research which is detailed in the manuscript. We will gladly complete this paragraph.*

3) The results chapter is generally well-designed but in places transgresses the boundary between genuine data presentation and interpretation, perhaps the most commonly found criticism in reviews these days. That could be rather easily solved by using a header such as Data "Presentation and Interpretation" and by restructuring the text in a manner that physically separates (paragraphs) more descriptive text from more interpretative text. Again, by this I aim for a better accessibility of the text for the nonspecialized reader. Clearly, chapter 3.5 is more of an interpretation than a genuine data reporting. Please consider.

**Authors' response:**

*We suggest to rename this part "Results and Interpretation". We agree to modify the structure of the text to try to separate more clearly interpretation from the results themselves, as suggested.*

4) Discussion: I was – in places – a bit confused about the manner in which the authors jump between lacustrine and marine micropearls. Could you do the reader the favour and commence the discussion with a paragraph explaining the reader how you structure the text? After all, the title of this paper refers to lacustrine and marine case examples but I do not see any chapter in the discussion specifically referring to lacustrine micropearls (but there is one dealing with marine examples actually referring to freshwater ones in the first paragraph)? I am aware of the fact that you have published on lacustrine examples before and that you place the lacustrine and marine findings into context but this all seems a bit unbalanced and would clearly benefit from some form of a better structure.

**Authors' response:**

*We definitely agree to write a first short paragraph in the discussion, explaining the structure of the text. Appart from that, we admit that our main idea when writing the discussion was to discuss the Tetraselmis genus as a whole, and we haven't been attentive to separate the marine from freshwater species. We suggest renaming part 4.1 "Marine and freshwater micropearls" as a first step. We could then distinguish clearly the marine micropearls from the freshwater ones by doing two separate paragraphs in part 4.1. For the other parts of the discussion (4.2 to 4.4), we do not think it is relevant to make that distinction, as they discuss the Tetraselmis genus in general, without reference to the species' habitats.*

5) Generally, I found the literature cited on ACC somewhat "classical". These are clearly landmark papers worth citing but a series of more recent studies dealing with thermodynamic and kinetic and biomineralization aspects on how and why organisms secrete or induce amorphous phases seems absent or so I think. I am happy to provide references should the authors wish so.

**Authors' response:**
*The various questions of the reviewers on this topic lead us to already plan to add the following references (see below) to the future revised manuscript. However, we thank the reviewer for his suggestion and we will also include the additionnal references he suggests to the revised ms.*

Albéric, M., Bertinetti, L., Zou, Z., Fratzl, P., Habraken, W., and Politi, Y. The crystallization of amorphous calcium carbonate is kinetically governed by ion impurities and water. Adv. Sci., 5, 1701000, 2018.

Aizenberg, J., Lambert, G., Weiner, S., and Addadi, L. Factors involved in the formation of amorphous and crystalline calcium carbonate: a study of an ascidian skeleton. J. Am. Chem. Soc., 124, 32-39, 2002.

Dupraz, C. , Reid, R.P., Braissant, O., Decho A.W., Norman, R.S., and Visscher, P.T. Processes of carbonate precipitation in modern microbial mats, Earth-Sci. Rev., 96, 141-162, doi:10.1016/j.earscirev.2008.10.005, 2009.

Levi-Kalisman, Y., Raz, S., Weiner, S., Addadi, L., and Sagi, I. Structural differences between biogenic amorphous calcium carbonate phases using X-ray absorption spectroscopy. Adv. Funct. Mater.,12, 43-48, 2002.

Littlewood, J.L., Shaw, S., Peacock, C.L. Mechanism of enhanced strontium uptake into calcite via an amorphous calcium carbonate (ACC) crystallisation pathway. Cryst. Growth Des., 17, 1214-1223, 2017.

Mass, T., Giuffre, A.J., Sun, C.-Y., Stifler, C.A., Frazier, M.J., Neder, M., Tamura, N., Stan, C.V., Marcus, M.A., and Gilbert, P.U.P.A. Amorphous calcium carbonate particles form coral skeletons. PNAS, 114,  E7670-E7678, 2017.

Politi, Y., Batchelor, D.R., Zaslansky, P., Chmelka, B.F., Weaver, J.C., Sagi, I., Weiner, S., Addadi, L. Role of magnesium ion in the stabilization of biogenic amorphous calcium carbonate: a structure-function investigation. Chem. Mater., 22, 161-166, 2010.

Rodriguez-Blanco J.D., Sand K.K. and Benning L.G. ACC and vaterite as intermediates in the solution-based crystallization of CaCO3. Chapter 5 in "New Perspectives on Mineral Nucleation and Growth", edited by Van Driessche A.E.S, Kellermeier M., Benning L.G. and Gebauer D. Springer, 2017.

6) The chapter, brief as it might be, that I really miss is one providing the reader with information about the wider significance of the amorphous calcium carbonate with respect to carbonate cycles and elemental cycling in these water bodies. Could you provide back on the envelope estimates about the volumes of material that are cycled here and the temporal constraints (seasonal patterns)?

**Authors' response:**
*Unfortunately, it is definitely too early to be able to estimate, even by back of the envelope calculations, the importance that this biomineralisation pathway might have in quantitative terms when the whole carbonate cycling in surface waters is considered. The timescales of*

*formation and the fate of the micropearls (dissolution and/or conversion to another mineral state) are yet unknown.*

*Nevertheless, we will add the following paragraph as an answer to the first part of the question: "ACC is an important actor in the biogenic carbonate cycle because it is a frequent precursor to calcite, as many organisms use ACC to build bio-minerals with superior properties (Albéric et al., 2018; Rodriguez-Blanco et al. 2017). For example, the precipitation of calcium carbonate in microbial mats, the Earth's earliest ecosystem, starts with an amorphous calcite gel (Dupraz et al., 2008), and the formation of ACC inside tissue could make coral skeletons less susceptible to ocean acidification (Mass et al., 2017)."*

7) Conclusions: Please don't come up with a paper written in 2010 (Raven and Knoll) and refer to something that was considered non-existent at that time. I agree, eight or so years seem not long ago but in our hyperactive research environment, this is actually a long time and significantly more accurate and recent findings have been published since then.

**Authors' response:**
*We agree that 2010 is not a very "recent" reference. Nevertheless, we have not found a more recent reference which lists all the known organisms having ACC intracellular inclusions. Additionally, we did not find any article mentioning a unicellular eukaryote organism known to produce intracellular ACC inclusions, apart from Tetraselmis. We would be very keen to read new articles on this subjects that we are not aware of.*

Summing up: My comments aim to improve an already very nice paper and I clearly encourage the journal Biogeosciences to consider publication of this work. My comments are on a conceptually high level given that a technical review is already available and given that I would like to see a wider readership making use of the science documented here.
I hope my comments are of use to the authors.
Sincerely yours, Adrian Immenhauser

---

## Author Comment (AC3) · 6 Jul 2018

**Reply to reviewer n°3**

**M. Alberic (Referee)**
marie.alberic@mpikg.mpg.de

The presence of intracellular amorphous calcium carbonate inclusions (called micropearls) has been identified for the first time in marine unicellular micro-algae (genus Tetraselmis). A wide range of marine species has been studied and compared to a fresh water species from the same genus. Careful and high quality structural and chemical investigations were performed, which allow characterizing the main structural features (shape, size, spatial localization of the micropearls and other cellular components) as well as chemical composition (Sr/Ca ratio) of the micropearls for each species. The results are discussed in terms of biomineralization processes, possible functions of the micropearls are proposed and bioremediation application highlighted. I believe this study is very interesting for the readership of Biogeosciences and the manuscript is very well written. The authors may address the following comments that mainly concern the organization of the results and a more advanced discussion part that could impact more fields as for example the "ACC stabilization" research community.

My only small concern is about the timescale of the biomineralization processes.
It will strengthen the paper if some scale can be provided.

**Authors' response:**
*For detailed answers to this point, please refer to questions 3, 5 and 12 (see detailed comments).*

Detailed comments

1) Introduction. The first sentence giving the definition of "micropearls" should be precised because this term was first proposed in the last study of the authors (Martignier 2017) for one genus (Tetraselmis). "Intracellular mineral inclusions of ACC" have been identified before in others species (in particular cyanobacteria) and were not called micropearls. Therefore, the genus should be stated and the reference (Martignier et al. 2017) added. In addition, previous studies (Couradeau et al 2012 and later ones Benzerara et al. 2014), should be cited even if they concern prokaryote organisms.

   **Authors' response:**
   *We thank the reviewer for this remark. As requested, in the revised version of the manuscript we will refer to our previous publication where the term micropearl was first coined. This article, however, was published four years after the first description of ACC in cyanobacteria and, thus, we do not rule out the possibility that the ACC inclusions produced by the*

*cyanobacteria could be also considered as micropearls. We will cite the publications about cyanobacteria in this part of the ms as suggested by the reviewer.*

2) Line 20. "two freshwater organisms", could mean either two individuals or two species: Only one is cited (cordiformis), what is the other one?

**Authors' response:**
*As answered to question 5 of Reviewer n°1:*
*We will replace the present sentence by the following: "*Until now, micropearls had been observed only in two freshwater species: the unicellular green alga *Tetraselmis cordiformis* (Chlorodendrophyceae, Chlorophyta) producing micropearls enriched in Sr and a second freshwater microorganism producing micropearls enriched in Ba, yet to be identified (Martignier et al., 2017)."*
*For the moment, the only thing we know about this second organism is its approximate size (20 microns) and that it has at least two flagella. We are currently trying different methods to investigate the question (isolation by cytometry, pipette subsampling).*

3) The presence of micropearls might depend on the time of observation of the cells. A time scale should be therefore indicated approximately, in order to make sure that it is the same for all culture cells. Would it be possible that micropearls belonging to different species could have different sizes, shapes, spatial localizations, Sr/Ca ratios just because of different time scales and not because they are from different species?

**Authors' response:**
*As explained in the Samples and Methods part (2.1): Samples for microscopic observation were prepared directly after the organisms' arrival in our laboratory".*
*At the time of this study, we did not have the necessary infrastructure or experience to maintain these algal cultures in satisfactory conditions on the long term. Therefore, as stated in paragraph 2.1, the present manuscript does not study the evolution of the cells with time, but describes an "instantaneous picture" of the micropearls in the cultures at the moment of their reception from the different algal culture collections. Giving a precise time scale for the different cultures is impossible, but these were all cultures with "mature cells" at the time of observation. So, the precision which we can add to the manuscript, is that "most cells in these cultures were mature at the time of observation".*
*Regarding the variability of the spatial localization (or patterns) of micropearl distribution in the cells, they don't seem to change during time (meaning that one species will not change pattern in the course of its life cycle). If it were the case, several different patterns should be observed simultaneously in the cultures (representing younger or older cells), which never occured. See also answer to question 1 of Reviewer n°1.*

*We can just add that our group recently obtained funds to finance a bi-disciplinary PhD student (microbiology / earth sciences) who is currently starting cultures and will work, amongst other aspects, on the study of the complete process of cell growth (and micropearl formation) from the start to the total maturation of the cells.*
*Preliminary analyses seem to confirm that the general pattern of micropearls localization seems to stay the same in one given species. Nevertheless, this study just started, the culture, analyses and sampling techniques still need to be improved and many more observations and analyses will be needed before these points can really be clarified.*

4) Is the rate growth of the micropearls known?

**Authors' response:**

*Regarding the growth rate of the micropearls, the only indication we have for the moment is theoretical: Thien et al. (2017) made an evaluation based on thermodynamic modelling of the formation of the micropearls: "If the size of the micropearls ranges between 0.3 and 2.5 µm, their potential growth time would then be between 0.6 and 72 days."*
*Reference:*
Thien, B., Martignier, A., Jaquet, J. M. and Filella, M.: Linking environmental observations and solid solution thermodynamic modeling: The case of Ba- and Sr-rich micropearls in Lake Geneva, Pure Appl. Chem., 89, 645–652, doi:10.1515/pac-2017-0205, 2017.

5) Does the compositional zonation (number of lines or spacing) could be a marker of time? or a marker of the different steps in the biomineralization process?

**Authors' response:**

*Compositional variations (e.g. oscillatory zoning) can occur generally in two different ways. Firstly, the composition of the growth medium is varying with time and this variation is documented in the zoning pattern. In this case, the growth zonation is temporally coupled to the variation of the external medium (marker of time) but often modified by variations in the growth rate. Such external control has to be ruled out here, since the culture cells grew from a medium of constant composition (no cyclic variations). Secondly, compositional variations in a solid can be induced by nonlinear dynamics involving a coupling between solid and fluid composition, and the oscillations are an example of chemical self-organisation in diffusion-reaction system (e.g. Liesegang rings). However, a biological control could influence the growth patterns, but is not yet identified.*

6) 2.3.The coating was gold, therefore the authors should state why carbon, nitrogen and oxygen where not taken into account in the semi-quantitative analyses.

**Authors' response:**

*Indeed, the Au peaks do not overlap with the peaks of these light elements. Nervertheless, we did not include C, N and O into the analyses because they are main components of organic matter and we have no means to distinguish how much of these elements comes from the micropearls and how much comes from the surrounding organic matter.*
*We wrote, in paragraph 3.4, that "It should be noted that the size of micropearls is close to or even below the resolution limit of the SEM-EDXS analysis technique. This means that the interaction volume of the electron beam with the sample is often larger than the micropearls themselves and that therefore the technique yields compositions that include the micropearl and the surrounding organic matter (…)".*

7) 2.5. Some EDXS have been done so the title should be changed accordingly

**Authors' response:**

*We thank the reviewer for spotting this error. We will change the title as suggested.*

8) 3. Results. In general the subtitles are not homogenous. If the authors choose to name the subtitles according to the techniques they use they should be more consistent, and therefore called 3.1 "SEM observation: : :" , and possibly put 3.2 and 3.3 together and call it TEM-EDXS, (because EDXS analysis are also reported in 3.2).

**Authors' response:**
*We understand the remark of the reviewer. We agree to change the subtitles for a more homogenous naming, as suggested by the reviewer. We would nevertheless prefer to keep paragraphs 3.2 and 3.3 separated. We consider TEM-EDXS mapping to produce sufficiently different results to TEM-EDXS analyses to justify separate paragraphs.*

9) 3.1. In Figure 1, the full name of the samples are not reported for d) and j) that have different strains. The different strains might be very similar but this should be specified.

**Authors' response:**
*We totally agree with this remark. We will complete the legend of this figure. The strain used for image (d) is cord-M_cc and the strain used for (j) is tetrath_ac. We will similarly indicate that the strain used for image (a) is chui_cc.*

10) In addition in Table 1 there is a mistake in the sample names for cui_sa and chui_cc.

**Authors' response:**
*For chui, the first line/last column should be chui_cc, and the second line/last column should be chui_sa. Regarding chui_cc, we also realised that the name of the strain is SAG 8-6 instead of SAG 8.6 as indicated. We will correct these mistakes.*

11) Line 11. "strains"or "species"?

**Authors' response:**
*At page 6, line 11, both terms would be correct. We agree to change to "species", which is more in accordance with the rest of the paragraph.*

12) p.7 line 1. The "problem" of the time of observation appears here, it is reported just for T. sriata, but the different organization in the different species could not be also related to different time scale? For T. Levis "the aggregate is missing" again, is this related to time?

**Authors' response:**
*Please refer to our answer to question n° 3 above and to question n°1 of Reviewer n°1.*
*As explained in our answer to question n°3, the present research project does not observe the evolution of the micropearls through the lifetime of a cell.  This manuscript simply states the presence of micropearls in 10 species of Tetraselmis (mature cells), with a given arrangement at the time of observation (time of arrival of the cultures in our lab). It is an "instantaneous picture".*
*We will add a sentence in the introduction or in the Results part (or both), stating this more clearly, in order to avoid misunderstanding.*
*Although, as detailed in the answer to question 1 of Reviewer 1, it appears clearly that the general pattern of micropearl distribution in the cell does not drastically change during time. As stated in the ms, the pattern of micropearls seems to be characteristic of a species.*

*Finally, to answer precisely the question: at p7, Line 1, the exact sentence we wrote is : "T. striata shows a similar central micropearl distribution, but the lateral points of the "trident" are absent (Fig. 1g), possibly due to poorly developed micropearls at the time of*

observation". *When we speek of "poorly developed micropearls", it does not refer to a "time problem", but states an apparent poor state of the micropearl development, which might be due to numerous factors. The new observations we provided for question 1 of Reviewer n°1 made us realise that this might be linked to the fact that this species was maintained on Agar, which seems to hinder the micropearl development. We will modify the text to include this new observation.*

13) Polyphosphate inclusions are not easily seen in Figure 1. Higher magnifications would be useful. Or relate to the TEM- EDXS observations? Did the authors observed EDXS signal from the P in these SEM images? Same for Iron-oxide minerals.

**Authors' response:**
*We agree that the reviewer's suggestions might indeed improve this figure, although we feel that the polyphosphate inclusions are large enough to be seen in Figure 1 without higher magnification, as they are larger than the micropearls. Therefore, we modified the figure by adding arrows pointing to the polyphosphate inclusions (P), as well as to the iron oxides (IO), in order to clearly identify them. See attached modified figure.*
*Regarding the composition of these items, we indeed performed EDXS analyses. We have attached a new figure, which could be inserted into the ms as a supplementary figure. It shows the EDXS analyses done in T. convolutae (Fig. 1c). This example shows raw EDXS analyses of polyphosphates, micropearls and iron oxides. Please note that the scale bar is 5 micrometers. Au, O and C were not taken into account for the semi-quantitative quantification (and these peaks stayed therefore in orange).*

[Figure]

[Figure]

14) Line 22 and 25 "organisms" or "species"?

**Authors' response:**

*Indeed, the word "species" might be more appropriate here than the word "organisms". Will modify the text accordingly.*

15) The authors cannot really state that "most flagellates do not produce micropearls" if they studied only two other species of one genus. They should be more careful, and maybe write instead of "most, : : : do not produce" "not all, : : : produce".

**Authors' response:**

*We understand the reviewer's concern. We agree to apply the suggested changes.*

16) 3.2. The choice of the samples for the FIB-sections is not clear. Why T. cordiformis from the culture was not considered? it would have been useful to compare with the natural environment one. It looks like the choice of the species was made in order to observe the compositional zonation. However, the compositional zonation of the different species in barely describe in the paper.

**Authors' response:**

*The choice of species for FIB-sections was indeed made to have the best chances of detecting possible compositional zonations in the micropearls of the marine species, similar to the ones already observed in the freshwater species. We agree that it would also be very interesting to have FIB-sections made in the cultured strains of T. cordiformis, as well as in other marine*

*species such as T. desikacharyi (where micropearls also contain small amounts of Ba). Nevertheless, FIB preparation is very time-consuming and costly. Therefore, for this publication, we reduced our choice to the three species that appeared to be the most promising ones.*

17) Figure 2. The red line is not visible in a black and white printed version, a dashed white line will be more useful. Higher magnification of individual micropearls like in Martignier et al. 2017 will be useful to better see the compositional zonation.

**Authors' response:**

*We agree to apply the suggested changes. Regarding the higher magnification images of individual micropearls, they are already presented in Fig. S4. We prefer to leave these higher magnification pictures in a separate supplementary figure as they are now, to avoid adding anymore element to Figure 2, which is already composed of many different elements.*

18) Fig. S4 shows higher magnifications, but it is difficult to see the zoning pattern. Is it because of the image quality? It looks like T. contracta does not show zoning pattern at all? (in Fig S4,"c)"is missing).

**Authors' response:**

*Images in Fig S4 are not taken at higher magnifications, but are details extracted from Fig 2 and 3. Due to the instability of the samples under the electron beam, repeated imaging (e.g. at higher magnification) is not possible. The HAADF signal is sensitive to changes in the mass-thickness (combination of mean atomic number, density and thickness of the sample). However, if compositions vary continuously or if sharp interfaces between zones are inclined to the electron beam, then the zoning will not be visible in HAADF images or EDXS maps.*
*T. contracta has a very low concentration of Sr (close to or below the detection limit) in TEM-EDXS. No zoning was detected here (this is stated in page 9 (lines 4-6)). While T. chui is clearly zoned, T. suecica appears almost unzoned, but with quite elevated Sr concentration (Sr/Ca close to 1). However, two additional thin zones close to the rim are visible, but they are too small to do meaningful TEM-EDXS analyses. We have changed the figure caption of Fig S4 to clarify this.*

***Figure S4: TEM-EDXS analyses of T. contracta, T. chui and T. suecica micropearls.***
*Cut-out of single micropearls (left) from STEM – HAADF images of the FIB section shown in Fig 2 and 3. The location of the EDXS analyses (right hand-side table) is indicated by the corresponding numbers. Results are normalized to 100 at%. O is calculated stoichiometrically based on the cation concentrations (with absorption correction for sample thickness). (a) The micropearl of T. chui is showing a clear zonation which is well documented in the TEM-EDXS results. (b) The micropearl of T. suesica appears almost unzoned, but with elevated Sr concentration (Sr/Ca close to 1). However, 2 additional thin zones close to the rim are visible, but they are too small to do meaningful TEM-EDXS analyses.(c) T. contracta has a very low concentration of Sr (close to or under the detection limit) in TEM-EDXS. No zoning was detected here. In contrast, a low but significant presence of K was detected. However its analysis n°1 does not fulfil carbonate stoichiometry, which may be due to the excess C from organic matter. Note that the calculation mode for the analyses presented in this figure differ from those presented in the rest of the manuscript, as C and O are included in the composition in order to perform a meaningful absorption correction.*

19) p.9 line 8. "the highly hydrated" state of the ACC should be speculate more carefully. Could it be that the water associated with organic molecules around or within(?) the ACC micropearls could lead to the "strong response under the electron beam"? When dehydration occurs, does ACC eventually crystallize into calcite or is it still stable? Because of the presence of the Sr ?

**Authors' response:**

*The presence of organics has not been formally established inside the micropearls because of the lack of suitable analytical techniques. The application of NanoSIMS to micropearls, reported in our previous paper (Martignier et al., 2017) revealed the presence of organic signatures (CN and S) at the periphery of the micropearls, but the relatively low resolution (8-15 um) prevented the mapping of OM inside the micropearls. Hence, at this stage, we can neither exclude nor assert the presence of OM inside the micropearls. What is obvious is the high reactivity of the micropearls under the EDS beam (see Fig. 3 in the reference above). This could be ascribed to the presence of OM associated with water or, alternatively, to hydrated ACC, or both. We have not tested the "exploded" micropearls after EDXS analysis for crystallinity, but we observed that their reactivity persists even after months of storage. This might be an indication that ACC occurs in micropearls as a stable, hydrated form. Beam damage was also observed in ACC microspheres precipitated in vitro and was explained by dehydration (Rodriguez-Blanco et al., 2008). Granted this, we propose to modify our phrasing thus: "As already pointed out previously (Martignier et al., 2016), the micropearls are extremely sensitive to the action of the electron beam, indicating a vaporization of some of its components: either organic matter associated with water, or water contained in the amorphous calcium carbonate (Rodriguez-Blanco et al., 2008), or both. This ACC seems to be rather stable, for the beam sensitivity persists after more than five months of storage of dry samples at room temperature."*

20) The unexpected stability of ACC in living systems is still highly debated. And to my knowledge not much studies so far reported the role of Sr on ACC stability, therefore, it will be worth discussing it in the discussion part.

**Authors' response:**

*Proposed modification for lines 21-25, p. 14:*
*"Non-biogenic ACC is unstable and will rapidly crystallize into calcite or aragonite (Addadi et al., 2003; Bots et al., 2012; Weiner and Addadi, 2011), often through the intermediate form of vaterite (Rodriguez-Blanco et al., 2011). In contrast, long-term stabilization of ACC implies the presence of mineral or organic additives (Aizenberg et al., 2002). Magnesium is known to play a key role in the stabilization of ACC by introducing a distortion in the host mineral structure (Politi et al., 2010). This might well be the case for the Tetraselmis-hosted micropearls, in which Mg content is around 2 mol%. Although the phosphate ion has also been reported to inhibit ACC crystallization (Albéric et al., 2018), it does not seem to be the case here, the phosphorus concentration of the micropearls being below the detection level of EDXS. Stabilization of ACC is also enhanced by certain proteins, polyphosphonates, citrates, amino acids (Levi-Kalisman et al., 2002; Addadi et al., 2003; Cam et al., 2015; Cartwright et al., 2012). The presence of these macromolecules inside the micropearls could be postulated by their observed sensitivity to beam damage. As for the possible role of Sr in the ACC long-term stability, we did not find in literature any reference thereof. However, in an in vitro experiment, Littlewood et al. (2017) found, in the presence of Mg, a correlation between added Sr and the reaction time to transform ACC into calcite (2 h to a maximum of 24 h)."*

21) line 14. Describe in which sense (number of lines, spacing: : :) the zoning pattern varies within one cell. Could it be related to different stage of biomineralization within one cell? The same for the different composition within one cell?

**Authors' response:**

*We agree to add a more detailed description of these variations, as observed in Fig. 2. Regarding the reason for these variations, these questions are almost impossible to answer given the current stage of knowledge. We know next to nothing about the dynamics of the micropearls, their formation and whether they can be dissolved or are only diluted to daughter cells during division. Intracellular calcium is required during the early stages of scale formation in the Golgi apparatus during cell division (which occurs in the dark period of the daily cycle). What controls the number and distribution of micropearls within a cell is also unknown.*
*Please also refer to our answer to question n°5.*

22) 3.3. line 22-27. The low magnification of Figure 3a and 3b does not allow properly visualizing the different cellular components, namely starch grains and chloroplasts. On which criteria the authors based the identification of these elements? Structural features? If so, higher magnifications of the areas presenting the chloroplasts and starch grains are needed. Do starch grains and chloroplasts can be characterized by specific chemical elements like PolyP and the scales? What about mitochondrial profiles? Are they also identified according to their shape? Higher magnification is then needed.

**Authors' response:**

*We can indeed provide zoom-in images of the chloroplasts, starch grains and mitochondrial profiles. We suggest to add them as a supplementary figure.*
*Organelles were tentatively identified based on both structural as well as positional information derived from previous transmission electron microscopy studies of T. cordiformis (Melkonian, 1979). There is one cup-shaped, reticulate chloroplast per cell. Except for its anterior open end, it is closely associated with the cell surface. It has been identified based on the presence of a smooth stroma and parallel arranged thylakoids (the latter usually in negative contrast). Starch grains are always located inside the chloroplast, otherwise they resemble micropearls (which are located outside the chloroplast) in overall shape and size, being however more irregular in their size distribution and revealing no structured core. Finally, there is only a single highly reticulated mitochondrion, consisting of an anastomosing network of tubules. When cross-sectioned these tubules give the impression of separate organelles. The mitochondrial profiles are predominantly located near the inner surface of the chloroplast.*

*Reference:*
*Melkonian, M.: An ultrastructural study of the flagellate* Tetraselmis cordiformis *stein (Chlorophyceae) with emphasis on the flagellar apparatus, Protoplasma, 98, 139–151, doi:10.1007/BF01676667, 1979*

23) 3.4. In Fig. 4, it would help the reader to indicate which species are from marine and fresh environment. Are this data coming from SEM-EDXS or TEM-EDXS or both?

**Authors' response:**

*We can indeed indicate which species are from a freshwater environment. We attach an accordingly modified version of Fig. 4: In Fig. 4a, the freshwater strains are distinguished by an asterisk following their name. In Fig. 4b, the freshwater strains are identified by a blue star in the center of the black circle in the graph.*
*All these EDXS analyses were carried out on SEM. We will add this information in the legend.*

[Figure]

24) Fig. S5 reports TEM-EDXS, and the section is about SEM-EDXS. No SEM-EDXS figures seem to be reported neither in the text nor in the supplementary info? Or Fig. S5 is a SEM-EDXS figure. Could the authors please clarify this point?

**Authors' response:**
*Fig. S5 is linked to section 3.3, not 3.4: (P11, line 11) "TEM-EDXS mapping provides compositional information improving the identification of the cellular constituents and organelles visible in the section (Fig. 3c and S5)."*
*It is true that we do mention again Fig. S5 in paragraph 3.4, but it is just to illustrate why some elements were not taken into account when calculating the composition of the micropearls, using the SEM-EDXS results. As Fig. S5 is not part of the results presented in paragraph 3.4, we do not think there is a need to change the text.*

25) 3.5. p13. Line 2. If the "overall composition of all culture media is rather similar" how could one study the influence of the composition of the media on the micropearls composition?

**Authors' response:**
*As illustrated in Fig. S6, although the overall composition of the culture media are rather similar, there are nevertheless clear differences. We will delete this first sentence of the paragraph, in order to avoid any misunderstanding.*

26) An interesting result is the one line 18, which would I think deserve more discussion in 4.1.

**Authors' response:**
*We thank the reviewer for this suggestion. We will modify the text p. 12, as from line 3:*
*"The Sr/Ca ratio seems to be influenced by several parameters, amongst which we identified the composition of the culture medium and the Sr concentrating capacity of each Tetraselmis species (dotted isolines in Fig.4b). The broad trend seen in this diagram could indicate a kind of adaptation of the ACC precipitation to the medium composition. However, a more relevant information is given by the enrichment factor (E factor), which can be ranked amongst species (Table S4), from low values (12-16) to more than 200. The reality of this ranking could be tested by cultivating the species in different media (eg. the convolutae group in ES and the tetrathele group in ASP-H) and comparing the new enrichment factor with the present values. The very high E factor for desikacharyi could be linked to distinctive morphological features (a six-layered theca, a novel flagellar hair subtype) not found in other strains of Tetraselmis (Marin et al., 1996)."*
Table S4 : Ranking of the Enrichment factor amongst species

| E factor | Strain | Medium | E factor | Environment |
|---|---|---|---|---|
| | cord_L | Lake Geneva | 12 | Freshwater |
| Low | convol_cc | ASP-H | 14 | Marine symbiotic |
| | contract_cc | ASP-H | 16 | Brackish |
| | subcord_sa | Porph Ag | 30 | Marine |
| Mdium | cord-F_cc | SFM | 33 | Freshwater |
| | chui_cc | ASP-H | 33 | Marine |
| | chui_sa | 1/2 SWEg Ag | 33 | Marine |
| | tetrath | ES | 42 | Brackish |
| High | striata_sa | SWES Ag | 43 | Marine |
| | cord-M_cc | Waris-H | 51 | Freshwater |
| Very High | desika_cc | ASP-H | 219 | Marine sand |

*Reference:*
Marin, B., Hoef-Emden, K. and Melkonian, M.: Light and electron microscope observations on *Tetraselmis desikacharyi* sp. nov.(Chlorodendrales, Chlorophyta), Nov. Hedwigia, 112, 5 461–475, 1996.

27) 4. Discussion. 4.2. p.14 line 18 to 25. Stabilization of ACC might also be achieved by inorganic ions (phosphates, magnesium, strontium?). The concentration of these ions might still be controlled by the organism. More recent papers about the stabilization of synthetic ACCs in presence of inorganic ions should be cited.The role of Strontium in the stabilization ACC should be further discussed by reporting the literature.

**Authors' response:**
*Please refer to our answer to question 19.*

28) To what refers "ACC in its "pure form"? this is rather vague, the authors could use the term "synthetic ACC with no additives" instead?

**Authors' response:**
*We will modfy the text according to the reviewer's suggestion.*

29) Moreover, synthetic ACC even if without inorganic or organic additives can be stable if it is stored in a desiccator for example. ACC in solution crystallizes indeed rapidly but not in air.

**Authors' response:**
*We would greatly appreciate to get the reference related to these statements. Our samples are not stored in a desiccator, but at normal "ambient conditions".*

30) 4.3. The spatial localization of the micropearls in a cell as well as the differences between cells of different species should be discussed. Does the specific localization close to the flagella have a role in swimming capacities? Center of gravity?

**Authors' response:**
*Unfortunately, it is impossible to answer these questions at the present stage of the research. For the moment, we have only been looking at dried cells with TEM or SEM and we still lack 3D views and sections of unaltered Tetraselmis cells (with preserved micropearls) to accurately locate the micropearls with respect to the various organelles and cell constituents. Work is planned to clarify these points using cryo TEM. But for the moment, as mentioned in the text (p. 15, lines 1-4), the possible role of the micropearls in Tetraselmis remains hypothetical.*

31) 5. Conclusion. p.16 line 6to 9 belongs to discussion not really to the conclusion

**Authors' response:**
*We suggest to modify this paragraph as follows:*
"Micropearls represent a new intracellular feature. This study shows that they can be clearly distinguished from other cellular constituents and are not randomly distributed in the cell. On the contrary, micropearls seem to be essentially located just under the cell wall and they draw a pattern which suggest to be characteristic for each species. Strong correlations hint that this might have a link with the species habitat.
It appears that, for most of the observed Tetraselmis species, the biomineralization process leading to the formation of micropearls enables a selective concentration of Sr."

---

## Editor Comment (EC1) · L.J. de Nooijer (Editor) · 1 Aug 2018

Dear Dr Martignier and co-workers,

I have read your replies and looked at your improved manuscript. I will -as quickly as I can- consult with (some of) the reviewers before accepting your paper for publication.

Sincerely,

Lennart

<hr>

---

## Author Response (AR1)

UNIVERSITÉ
DE GENÈVE

FACULTÉ DES SCIENCES

Department of Earth Sciences
13, rue des Maraîchers
1205 Geneva

**Agathe Martignier**

Phone: +41 22 379 3164
Fax : +41 22 379 32 10
Agathe.martignier@unige.ch
http://cms.unige.ch/sciences/terre

**L.J. de Nooijer (Editor)**

Geneva, 5th of September 2018

Dear Sir,

Following your report, we submit here the revised version of our manuscript:

**Marine and freshwater micropearls:
Novel biomineralization process is widespread in the genus *Tetraselmis* (Chlorophyta)**

by Agathe Martignier and co-authors, which we had submitted for publication as a research article in *Biogeosciences*.

Following your advice, we have implemented all the modifications and changes, which we had mentioned in the "Answers to the Reviewers" documents, previously published on the "Interactive Discussion" page of the *Biogeosciences* website. We are grateful for the fact that this version of our manuscript has definitely been improved, thanks to the reviewers' questions and suggestions.

This letter includes, as an attachment, a version of our revised manuscript including the track changes.

Respectfully Yours,

Agathe Martignier          (on behalf of all co-authors)

/

[revised manuscript text omitted]

---

## Author Response (AR2)

UNIVERSITÉ
DE GENÈVE

**FACULTÉ DES SCIENCES**

Department of Earth Sciences
13, rue des Maraîchers
1205 Geneva

**Agathe Martignier**

Phone: +41 22 379 3164
Fax : +41 22 379 32 10
Agathe.martignier@unige.ch
http://cms.unige.ch/sciences/terre/

**L.J. de Nooijer (Editor)**

Geneva, 17th of October 2018

Dear Sir,

Thank you very much for your positive decision regarding our manuscript. We have applied to our manuscript (**Marine and freshwater micropearls: Novel biomineralization process is widespread in the genus *Tetraselmis* (Chlorophyta)**) the two last text modifications which you have suggested**.**

Attached to this letter, you will find a correspondingly revised version of our manuscript, including the track changes of the modifications (see part 4.2 and legend of figure1).

Respectfully yours,

Agathe Martignier          (on behalf of all co-authors)

[revised manuscript text omitted]